



# SPEAD 1.0 – A model for Simulating Plankton Evolution with Adaptive Dynamics in a two-trait continuous fitness landscape applied to the Sargasso Sea

Guillaume Le Gland[1], Sergio M. Vallina[2], S. Lan Smith[3], and Pedro Cermeño[1]

[1]Institut de Ciències del Mar (CSIC), Passeig Marítim de la Barceloneta 37-49, 08003 Barcelona, Spain
[2]Spanish Institute of Oceanography (IEO), Ave Principe de Asturias 70 bis, 33212 Gijon, Spain
[3]Earth SURFACE Research Center, Research Institute for Global Change, JAMSTEC, Yokosuka, Japan

*Correspondence to:* Guillaume Le Gland (legland@icm.csic.es)

**Abstract.**

Diversity plays a key role in the adaptive capacities of marine ecosystems to environmental changes. However, modeling phytoplankton trait diversity remains challenging due to the strength of the competitive exclusion of sub-optimal phenotypes. Trait diffusion (TD) is a recently developed approach to sustain diversity in plankton models by allowing the evolution of

functional traits at ecological timescales.

In this study, we present a model for Simulating Plankton Evolution with Adaptive Dynamics (SPEAD), where phytoplankton phenotypes characterized by two traits, nitrogen half-saturation constant and optimal temperature, can mutate at each generation using the TD mechanism. SPEAD does not resolve the different phenotypes as discrete entities, computing instead six aggregate properties: total phytoplankton biomass, mean value of each trait, trait variances, and inter-trait covariance of a single

population in a continuous trait space. Therefore SPEAD resolves the dynamics of the population's continuous trait distribution by solving its statistical moments, where the variances of trait values represent the diversity of ecotypes. The ecological model is coupled to a vertically-resolved (1D) physical environment, and therefore the adaptive dynamics of the simulated phytoplankton population are driven by seasonal variations in vertical mixing, nutrient concentration, water temperature, and solar irradiance. The simulated bulk properties are validated by observations from BATS in the Sargasso Sea.

We find that moderate mutation rates sustain trait diversity at decadal timescales and soften the almost total inter-trait correlation induced by the environment alone, without reducing the annual primary production or promoting permanently maladapted phenotypes, as occur with high mutation rates. As a way to evaluate the performance of the continuous-trait approximation, we also compare the solutions of SPEAD to the solutions of a classical discrete entities approach, both approaches including TD as a mechanism to sustain trait variance. We only find minor discrepancies between the continuous model SPEAD and the

discrete model, the computational cost of SPEAD being lower by two orders of magnitude. Therefore SPEAD should be an ideal eco-evolutionary plankton model to be coupled to a general circulation model (GCM) at the global ocean.



## 1 Introduction

Phytoplankton are a polyphyletic group of single-cell primary producers widespread in aquatic environments. Despite accounting for only 1% of the global photosynthetic biomass, they perform more than 45% of Earth's net primary production (Field et al., 1998; Falkowski et al., 2004). They are the basis of all oceanic food webs and play key roles in biogeochemical cycles (Falkowski, 2012). In particular, they have a large impact on global climate through the export of detritic carbon from the surface to the ocean interior, sequestrating carbon in the deep ocean for timescales from a few years to more than a millenium, depending on the depth they reach (DeVries and Primeau, 2011; DeVries et al., 2012). This process, called the "biological

carbon pump", regulates the concentration of carbon dioxide in the atmosphere (Volk and Hoffert, 1985; Falkowski et al., 1998).

Phytoplankton are highly diverse and live in many different environments. They differ in their ecological interactions and the processes through which they mediate biogeochemical cycles. For instance, some species can fix atmospheric nitrogen and enrich oligotrophic regions, some produce ballast minerals (mainly silica and calcium carbonate) and sink faster to the

deep ocean, some promote the formation of clouds by producing dimethylsulfide, and others are mixotrophic, being able to both photosynthesize and feed on organic sources (Le Quéré et al., 2005). Most species are denser than seawater and eventually sink but some are buoyant (Lännergren, 1979; Villareal, 1988). Phytoplankton size ranges from less than 1 $\mu$m for cyanobacteria like *Prochlorococcus* (Chisholm et al., 1988) to more than 1 mm for the giant diatom *Ethmodiscus rex* (Swift, 1973; Villareal and Carpenter, 1994). Their half-saturation constants for the main limiting nutrients range over three orders

of magnitude (Edwards et al., 2012), reflecting adaptation to different nutrient supply levels. They are also adapted to very different temperatures: some diatoms can grow within sea ice (Ackley and Sullivan, 1994), whereas some hyperthermophilic cyanobacteria can grow at up to 75°C in hot springs (Castenholz, 1969). Most oceanic species have an optimal temperature for growth between 0 and 35°C (Thomas et al., 2012). Even within the same species or genus, wide variability has been observed for key traits such as iron requirement (Strzepek and Harrison, 2004), light requirement (Biller et al., 2015) and resistance to

predation (Yoshida et al., 2004). Given that any change in the abundance and composition of phytoplankton has far-reaching consequences for other organisms and for the Earth's climate, it is important to understand the factors driving the dynamics of such communities.

Numerical modelling studies can address this issue by finding the mechanistic equations and parameters that most correctly account for the observations, and thereby provide invaluable insights into the general rules controlling ecosystems. Models

can also be used to make predictions of how phytoplankton will impact or be impacted by future environmental changes (Norberg et al., 2012; Irwin et al., 2015). Mathematical models of phytoplankton growth as a function of nutrient concentration, temperature and radiation have been developed since the 1940's (Riley, 1946; Steele, 1958; Riley, 1965), leading to the now common NPZD ("Nutrient, Phytoplankton, Zooplankton, Detritus") models representing one or several compartments of nutrients, phytoplankton, zooplankton and detritus (Fasham et al., 1990). Since the early 1990's (Maier-Reimer, 1993), the

increase in computational power allowed biogeochemical models to be fully coupled with ocean circulation (Aumont et al., 2003; Follows et al., 2007). However, representing all the phytoplankton diversity in models is neither feasible nor desirable.





The computational cost would be high, and even if computationally feasible, the existing observations would not suffice to constrain the many free parameters.

Instead, models account for biodiversity through a few key traits representing physiological characteristics or adaptation to different environments. The most widely investigated phytoplankton traits are cell size, nutrient niche, optimal temperature, optimal irradiance and resistance to predation. Some trait-based models divide the phytoplankton community into discrete entities or "boxes" with different traits. The boxes can be as simple as diatoms and small phytoplankton groups  (Aumont

et al., 2015), with diatoms having higher nutrient concentration niches, or include more complex divisions into functional groups  (Baretta et al., 1995; Le Quéré et al., 2005; Follows et al., 2007). The other approach, which further reduces the number of equations while still allowing communities to adapt to changes in their environments, is to consider one or several continuously distributed traits and to compute only the dynamics of aggregate properties, such as community biomass, mean trait values and trait variances  (Wirtz and Eckhardt, 1996; Norberg et al., 2001; Bruggeman and Kooijman, 2007; Merico et al.,

2009; Acevedo-Trejos et al., 2016; Smith et al., 2016; Chen and Smith, 2018). In this method trait variance can be used as a quantitative index of biodiversity. A community with a higher trait variance is considered to be more diverse because it has a wider spread and more even distribution of trait values  (Li, 1997), although it does not necessarily have a higher number of taxonomic species ("richness")  (Vallina et al., 2017).

One weakness induced by the simplification of phytoplankton communities in both the aggregate and discrete models,

however, is that competitive exclusion  (Hardin, 1960; Hutchinson, 1961) often leads to a collapse of the modeled diversity (Merico et al., 2009) unless trait variance is imposed  (Norberg et al., 2012; Wirtz, 2013) or a mechanism is added explicitly to sustain it. One way to sustain biodiversity is through immigration from a distant community  (Norberg et al., 2001; Savage et al., 2007). Yet, immigration does not explain the diversity observed in closed laboratory experiments, including continuous cultures  (Fussmann et al., 2007; Kinnison and Hairston, 2007; Beardmore et al., 2011). Biodiversity can also be sustained

by viruses  (Thingstad and Lignell, 1997) or predators  (Murdoch, 1969; Kiørboe et al., 1996) if they specialize on a narrow range of preys or switch their preference to the most common phytoplankton species. This is the idea behind the "Kill The Winner" theory  (Thingstad, 2000; Vallina et al., 2014b), where predation concentrating more so on the most dominant species maintains diversity, because then each prey species persists at the abundance where the predation rate equals its growth rate.

An alternative approach recently introduced to sustain diversity in models is to allow the simulated phytoplankton to mutate

their physiological traits  (Kremer and Klausmeier, 2013; Merico et al., 2014). Due to their short generation times of around 1 day  (Marañon et al., 2013), phytoplankton are known to evolve at the timescale of a few years  (Schlüter et al., 2016). For phytoplankton, the "ecological" timescales, featuring successions of dominant species in reaction to changes in the environment and selection of the fittest, overlap with the "evolutionary" timescales, where species can also evolve genetically to adapt to their new environment  (Irwin et al., 2015). As far as we know, the first aggregate phytoplankton model allowing a phyto-

plankton trait to randomly mutate through subsequent generations, before being selected by the environment, was developed by  Merico et al. (2014). They called their scheme "Trait diffusion" (TD), where "diffusion" is a mathematical term referring to the spreading of a property, in this case the trait value, not to physical transport. Trait diffusion of a single physiological trait was recently introduced in a model coupled with oceanic circulation  (Chen and Smith, 2018). Upgrading the trait dif-





fusion framework to several traits requires more complex equations and the introduction of a new class of state variables: the
covariances between traits. However, multi-trait models are more realistic and conceptual modeling studies have shown that
the dynamics of correlated traits sometimes differ from those of single-trait models (Savage et al., 2007).

Here we present a new aggregate phytoplankton model called SPEAD (Simulating Plankton Evolution with Adaptive
Dynamics), an eco-evolutionary model using the trait diffusion framework for two key phytoplankton traits: nitrogen half-
saturation constant and optimal temperature for growth. Our model is based on a NPZD model (Vallina et al., 2014a, 2017),
where the phytoplankton compartment is represented by the community biomass, mean trait values, trait variances and covari-
ance. SPEAD is embedded in a 1D (water column) physical setting simulating the Sargasso Sea using data from the Bermuda
Atlantic Time-series Studies (BATS). We chose the 1D rather than a simpler 0D setting because vertical turbulent diffusion
(not to be confused with trait diffusion) is the main source of covariance by mixing communities from different depths. Since
the trait diffusion equations can easily be adapted to a discrete model, we have also built a discrete version of SPEAD where
the phytoplankton community consists of 625 different phenotypes (i.e. 25 half-saturation constants and 25 optimal temper-
atures), each characterized by its own fixed set of trait values. The discrete version is more intuitive and easier to program,
and provides a useful control experiment. SPEAD is intended as a prototype to be coupled later with 3D general circulation
models. Its equations for mean trait, trait variance and covariance can be used as a starting point to build more comprehensive
trait-based models, with or without mutations. In particular, we plan to add optimal irradiance as a third trait in the near future
for a more realistic description of phytoplankton distributions over depth.

In the following sections, we first describe our ecological model, the differential equations controlling the growth of phyto-
plankton and the adaptive evolution of their trait distribution, as well as the physical model setting. Then, we present the model
outputs. In order to validate SPEAD and to highlight its novelties, we will focus on answering the following four questions: 1)
How well does SPEAD represent the bulk properties of phytoplankton communities observed in the Sargasso Sea? 2) Do the ag-
gregate and discrete approaches agree? 3) How are phytoplankton dynamics changed by the value of the mutation rate? 4) Can
the mean value and variance of each trait be represented independently by a 1-trait model where only nitrogen half-saturation
or optimal temperature varies between phenotypes? Finally, we discuss the reach of our modeling framework, focusing on three
aspects: the performance of aggregate models, the choice of phytoplankton traits, and the relationship between trait diffusion
and evolution.

## 2 Methods

### 2.1 A phytoplankton community model with 2 traits

Our phytoplankton community model SPEAD extends an existing nitrogen-based NPZD model (Vallina et al., 2017). Nitrogen
is partitioned into four pools, all expressed in millimoles of nitrogen per cubic meter (mmolN m$^{-3}$): phytoplankton ($P$ in the
equations), zooplankton ($Z$), Dissolved Inorganic Nitrogen or "DIN" ($N$) and Particulate Organic Nitrogen or "PON" ($D$
as in "detritus"). Phytoplankton increases its biomass by taking up DIN ($U_p$). Zooplankton increases its biomass by grazing
phytoplankton ($G_z$). The non-predation mortalities of phytoplankton ($M_p$) and zooplankton ($M_z$) and the nitrogen exudation





by zooplankton ($E_z$) are divided between DIN and PON. Given that nitrogen is the limiting nutrient for phytoplankton growth, we do not consider nitrogen exudation by phytoplankton. $\omega_p$, $\omega_z$ and $\epsilon_z$ are constants representing the respective proportions of $M_p$, $M_z$ and $E_z$ going to DIN. PON is remineralized to DIN ($M_d$). The fluxes from one pool to another are controlled by

the pool concentrations and by two environmental forcings: temperature ($T$, in °C) and Photosynthetically Available Radiation or "PAR" ($I$, in W m$^{-2}$). The main state variables of the model and their relationships are shown in Table 1 and Fig. 1a and the expressions of the fluxes are given by the following equations, with their dependencies:

$$\frac{dP}{dt} = U_p(P,N,T,I) - M_p(P,T) - G_z(P,Z,T) \tag{1}$$

$$\frac{dZ}{dt} = G_z(P,Z,T) - E_z(P,Z,T) - M_z(Z,T) \tag{2}$$

$$\frac{dN}{dt} = \epsilon_z E_z(P,Z,T) + \omega_p M_p(P,T) + \omega_z M_z(Z,T) + M_d(D,T) - U_p(P,N,T,I) \tag{3}$$

$$\frac{dD}{dt} = (1-\epsilon_z)E_z(P,Z,T) + (1-\omega_p)M_p(P,T) + (1-\omega_z)M_z(Z,T) - M_d(D,T) \tag{4}$$

Zooplankton, DIN and PON are generic pools, characterized by a single variable: their concentration. Phytoplankton and zooplankton mortality, zooplankton exudation, grazing and the particle remineralization rate have simple expressions as a function of the nitrogen pool concentrations and temperature:

$$G_z(P,Z,T) = g_0 e^{\alpha_h(T-T_0)} \frac{P^2}{P^2 + K_p^2} Z \tag{5}$$

$$E_z(P,Z,T) = (1-\beta_z)G_z(P,Z,T) \tag{6}$$

$$M_z(Z,T) = \psi_z e^{\alpha_h(T-T_0)} Z^2 \tag{7}$$

$$M_p(P,T) = \psi_p e^{\alpha_h(T-T_0)} P \tag{8}$$

$$M_d(T) = \psi_d e^{\alpha_h(T-T_0)} D \tag{9}$$

The constants appearing in this equation ($\alpha_h$, $T_0$, $K_p$, $\beta_z$, $\psi_z$, $\psi_p$ and $\psi_d$) are described in Table 2. Zooplankton mortality depends on the square of zooplankton concentration in order to prevent an explosion of zooplankton concentration. This stabilizing quadratic mortality term represents consumption by animals higher on the trophic chain, which is expected to increase faster than a linear function of Z biomass. Grazing is formulated as a Holling type III function (Holling, 1959) of phytoplankton concentration, with a niche at low concentrations to prevent the whole phytoplankton community from going

extinct, even when they have very low growth rates. Grazing, mortality and remineralization are considered as heterotrophic processes and as such increase exponentially with temperature. The exponential factor is $\alpha_h = 0.092$ °C$^{-1}$. This is equivalent to multiplying the speed of all these processes by 2.5 when the temperature increases by 10 °C, as in a "Q10 = 2.5" formulation. This value of Q10 is close to measured values for zooplankton grazing (Hansen et al., 1997) and to the theoretical predictions of the metabolic theory of ecology for respiration (Gillooly et al., 2001; Allen et al., 2005).

On the contrary, the phytoplankton pool is composed of diverse organisms responding to environmental conditions in different ways. The diversity of phytoplankton is represented by variations in the values of two traits: the logarithm of half-saturation constant for nitrogen uptake ($x$) and the optimal temperature for growth ($y$). From now on, we will refer to each set ($x$,$y$) of trait





**Table 1.** State variables of the ecosystem model

| Symbol | Description | Unit |
|--------|-------------|------|
| | Prognostic variables of the aggregate model | |
| $P$ | Phytoplankton concentration | $\text{mmolN}\,\text{m}^{-3}$ |
| $\overline{x}$ | Mean nitrogen half-saturation logarithm (trait 1) | – |
| $\overline{y}$ | Mean optimal temperature (trait 2) | $^\circ\text{C}$ |
| $V_x$ | Half-saturation logarithm variance | – |
| $V_y$ | Optimal temperature variance | $^\circ\text{C}^2$ |
| $C_{xy}$ | Inter-trait covariance | $^\circ\text{C}$ |
| | Prognostic variables of the multi-phenotype model | |
| $P_{ij}$ | Concentration of phytoplankton phenotype i,j | $\text{mmolN}\,\text{m}^{-3}$ |
| | Prognostic variables common to both models | |
| $Z$ | Zooplankton concentration | $\text{mmolN}\,\text{m}^{-3}$ |
| $N$ | Dissolved Inorganic Nitrogen (DIN) concentration | $\text{mmolN}\,\text{m}^{-3}$ |
| $D$ | Particulate Organic Nitrogen (PON) concentration | $\text{mmolN}\,\text{m}^{-3}$ |
| | Diagnostic variables related to trait | |
| $\sigma_x$ | Standard deviation of half-saturation logarithm | – |
| $\sigma_y$ | Standard deviation of optimal temperature | $^\circ\text{C}$ |
| $R_{xy}$ | Inter-trait correlation | – |
| | Other diagnostic variables | |
| $Chl$ | Chlorophyll a concentration | $\text{mgCHL}\,\text{m}^{-3}$ |
| $PP$ | Primary production | $\text{mgC}\,\text{m}^{-3}\,\text{d}^{-1}$ |
| | Environment variables | |
| $T$ | Temperature | $^\circ\text{C}$ |
| $I$ | Photosynthetically available radiation (PAR) | $\text{W}\,\text{m}^{-2}$ |
| $K_z$ | Vertical diffusivity | $\text{m}^2.\text{d}^{-1}$ |





**Table 2.** Parameters of the ecosystem model

| Symbol | Description | Value | Unit |
|---|---|---|---|
| | *Phytoplankton parameters* | | |
| $T_0$ | Reference temperature | 20 | $^\circ$C |
| $u_p^0$ | Phytoplankton maximum uptake rate for $x = 0$ and $y = T_0$ | 1.1 | d$^{-1}$ |
| $\Delta T$ | Difference between optimal and maximal temperature | 5 | $^\circ$C |
| $\epsilon_p$ | Phytoplankton exudation fraction going to DIN | 1/3 | – |
| $\psi_p$ | Phytoplankton mortality rate at 20 $^\circ$C | 0.05 | d$^{-1}$ |
| $\omega_p$ | Phytoplankton mortality fraction going to DIN | 1/4 | – |
| $I_0$ | Phytoplankton optimal irradiance | 25 | W m$^{-2}$ |
| $\chi$ | Phytoplankton photoinhibition factor | 12 | – |
| $\alpha_a$ | Temperature dependence factor for autotrophic processes | 0.056 | $^\circ$C$^{-1}$ |
| | Speed multiplied by 1.75 ("Q10") with a temperature increase of 10 $^\circ$C | | |
| $\nu_x$ | Half-saturation diffusivity parameter | $10^{-5}$ – 0.1 | – |
| $\nu_y$ | Optimal temperature diffusivity parameter | $10^{-4}$ – 1 | $^\circ$C$^2$ |
| | *Other ecological parameters* | | |
| $g_0$ | Zooplankton maximum grazing rate at 20 $^\circ$C | 1.5 | d$^{-1}$ |
| $K_p$ | Half-saturation for grazing | 0.4 | mmolN m$^{-3}$ |
| $\beta_z$ | Zooplankton assimilation efficiency | 0.4 | – |
| $\epsilon_z$ | Zooplankton exudation fraction going to DIN | 1/3 | – |
| $\psi_z$ | Zooplankton (quadratic) mortality rate at 20 $^\circ$C | 0.25 | (mmolN m$^{-3}$)$^{-1}$ d$^{-1}$ |
| $\omega_z$ | Zooplankton mortality fraction going to DIN | 1/4 | – |
| $\psi_d$ | PON remineralization rate at 20 $^\circ$C | 0.1 | d$^{-1}$ |
| $w$ | PON sinking speed | 1.2 | m d$^{-1}$ |
| $k_w$ | PAR vertical attenuation | 0.04 | m$^{-1}$ |
| $\alpha_h$ | Temperature dependence factor for heterotrophic processes | 0.092 | $^\circ$C$^{-1}$ |
| | Speed multiplied by 2.5 ("Q10") with a temperature increase of 10 $^\circ$C | | |
| | *Numerical parameters* | | |
| $n_x$ | Number of half-saturation values (discrete model) | 25 | – |
| $n_y$ | Number of optimal temperature values (discrete model) | 25 | – |
| $x_{min}$ | Minimum half-saturation logarithm (discrete model) | -2.5 | – |
| $x_{max}$ | Maximum half-saturation logarithm (discrete model) | +1.5 | – |
| $y_{min}$ | Minimum optimal temperature (discrete model) | 18 | $^\circ$C |
| $y_{max}$ | Maximum optimal temperature (discrete model) | 30 | $^\circ$C |
| $z_{max}$ | Maximum model depth | 200 | m |



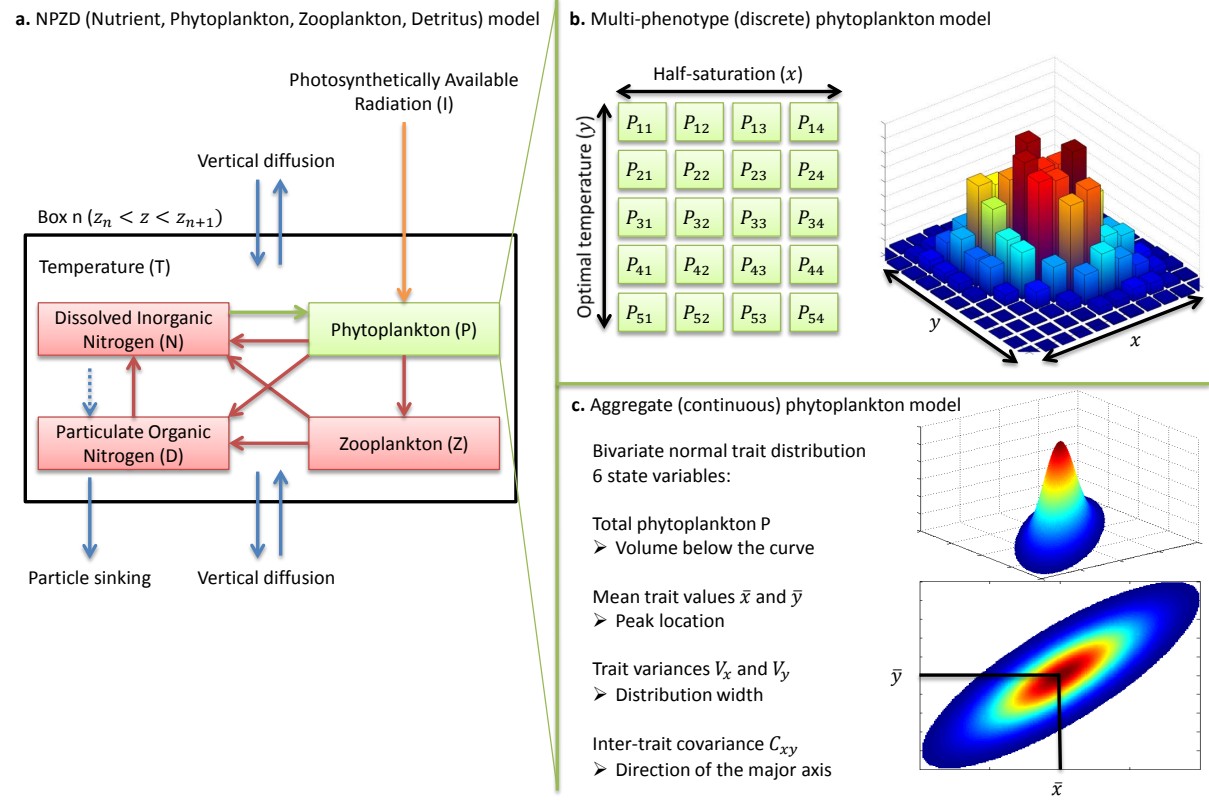

**Figure 1.** NPZD (Nutrient, Phytoplankton, Zooplankton, Detritus) model within its physical setting (a). The phytoplankton pool is represented by a discrete set of species with different traits (b) or by moments of the trait distributions, assuming a bivariate normal distribution (c). Colors in b and c represent phytoplankton concentration.

values as a "phenotype". Nutrient uptake by phytoplankton depends on the trait distribution. The bivariate trait distribution is

5   represented by a density $p(x, y)$ (in mmolN m$^{-3}$ °C$^{-1}$) so that the biomass of phytoplankton (in mmolN m$^{-3}$) with trait values between $x_1$ and $x_2$ and between $y_1$ and $y_2$ is equal to $\int_{x_1}^{x_2} \int_{y_1}^{y_2} p(x, y)dxdy$ and by extension the total phytoplankton biomass P is equal to the density integrated over the whole trait domain. Any phenotype has its own uptake rate $u_p(x, y)$. The uptake rate is the product of a constant ($u_p^0$) and three growth factors: a nutrient factor ($\gamma_n(N, x)$), a temperature factor ($\gamma_T(T, y)$) and an irradiance factor ($\gamma_i(I)$). Two of these factors, $\gamma_n(N, x)$ and $\gamma_i(I)$, represent limitations by resources. The third factor, $\gamma_T(T, y)$, represents the kinetic effect of temperature on growth. In this study, we use the Monod approach  (Monod, 1949), so that cells do not store nutrients and the uptake rate is equal to the reproduction or growth rate. Since phytoplankton are unicellular and we do not consider changes in their cell volumes, we will use the words "growth" and "reproduction" interchangeably.

All phenotypes share the same rates of mortality and grazing, respectively.





The last term in the equation of trait density (Eq. 10) is trait diffusion, as defined by Merico et al. (2014). Trait diffusion represents the fact that offspring can exhibit different trait values than their parents, due to mutations or otherwise heritable plasticity. In our numerical model, we assume only that these mutations are heritable and random. They can represent genetic mutations as well as other, e.g. epigenetic, phenotypic plasticity. We assume that mutations on $x$ and $y$ are independent of each

other. In the limit of small but frequent mutations, stochasticity can be neglected (Dieckmann and Law, 1996; Champagnat et al., 2006) and this process can be represented as a deterministic diffusion, depending on diffusivity parameters $\nu_x$ and $\nu_y$ and on the second derivatives of the growth rate ($u_p(x,y)$) relative to each trait respectively. Note that in TD, "diffusion" is a mathematical term referring to the spreading of a property, in our case trait values, not to a physical mixing process. It should therefore not be confused with vertical turbulent diffusion, which is also present in our model (see 2.3.). To avoid ambiguity,

from now on, we will refer to the trait diffusivity parameters as "mutation rates". $\nu_x$ and $\nu_y$ are mutation rates per generation, not per unit of time, therefore time does not appear in their units. They have the same units as trait variances. The derivation of the trait diffusion term is explained in Appendix A. The differential equations followed by a given phenotype (x,y) are:

$$\frac{\partial p(x,y,t)}{\partial t} = \left[ u_p(N,T,I,x,y) - \frac{M_p(P,T)}{P} - \frac{G_z(P,Z,T)}{P} \right] p(x,y,t) + \nu_x \frac{\partial^2 (u_p \cdot p)}{\partial x^2} + \nu_y \frac{\partial^2 (u_p \cdot p)}{\partial y^2} \tag{10}$$

$$u_p(N,T,I,x,y) = u_p^0 \gamma_n(N,x) \gamma_T(T,y) \gamma_i(I) \tag{11}$$

Like all biodiversity models, SPEAD must not allow a phenotype to outcompete all other phenotypes in all environments, because any such Darwinian demon would drive all its sub-optimal competitors to extinction and trait variance would collapse to zero. In order to make competition for nutrients possible, we have defined two uptake traits so that either low or high values are advantageous in certain environments and disadvantageous in others. The shape of the two trade-offs and the three growth factors are presented in Fig. 2.

The first trait allowed to mutate in SPEAD is the half-saturation constant that controls the nutrient limitation factor $\gamma_n(N,x)$. The half-saturation constant can be linked to the well-known trade-off between the affinity for a nutrient and the maximum uptake rate, also known as the "gleaner-opportunist" trade-off (Frederickson and Stephanopoulos, 1981). The biomass-specific nitrogen uptake rate $u_p(N,T,I,x,y)$ of a given phenotype is proportional to its affinity for nitrogen $f_p$ (in d$^{-1}$ mmol$^{-1}$ m$^3$) at low nitrogen concentration and reaches the maximum uptake rate $u_p^\infty$ (in d$^{-1}$) in nutrient-replete environments. The uptake

rate follows a Michaelis-Menten function of nutrient concentration:

$$u_p(N,T,I,x,y) = u_p^\infty(T,I,x,y) \frac{N}{\frac{u_p^\infty(T,I,x,y)}{f_p(T,I,x,y)} + N} \tag{12}$$

$$u_p^\infty(T,I,x,y) = \lim_{N \to \infty} u_p(N,T,I,x,y) = u_p^0 \gamma_T(T,y) \gamma_i(I) \lim_{N \to \infty} \gamma_n(N,x) \tag{13}$$

$$f_p(T,I,x,y) = \lim_{N \to 0} \frac{u_p(N,T,I,x,y)}{N} = u_p^0 \gamma_T(T,y) \gamma_i(I) \lim_{N \to 0} \frac{\gamma_n(N,x)}{N} \tag{14}$$

Phenotypes that specialize in taking up the few available nutrients at low concentrations (high $f_p$) are called "gleaners", whereas those that specialize in taking up nutrients quickly at saturating nutrient concentrations (high $u_p^\infty$) are called "opportunists". We assume that the product $f_p u_p^\infty$ is independent of $x$ (Meyer et al., 2015; Vallina et al., 2017), a relation that defines



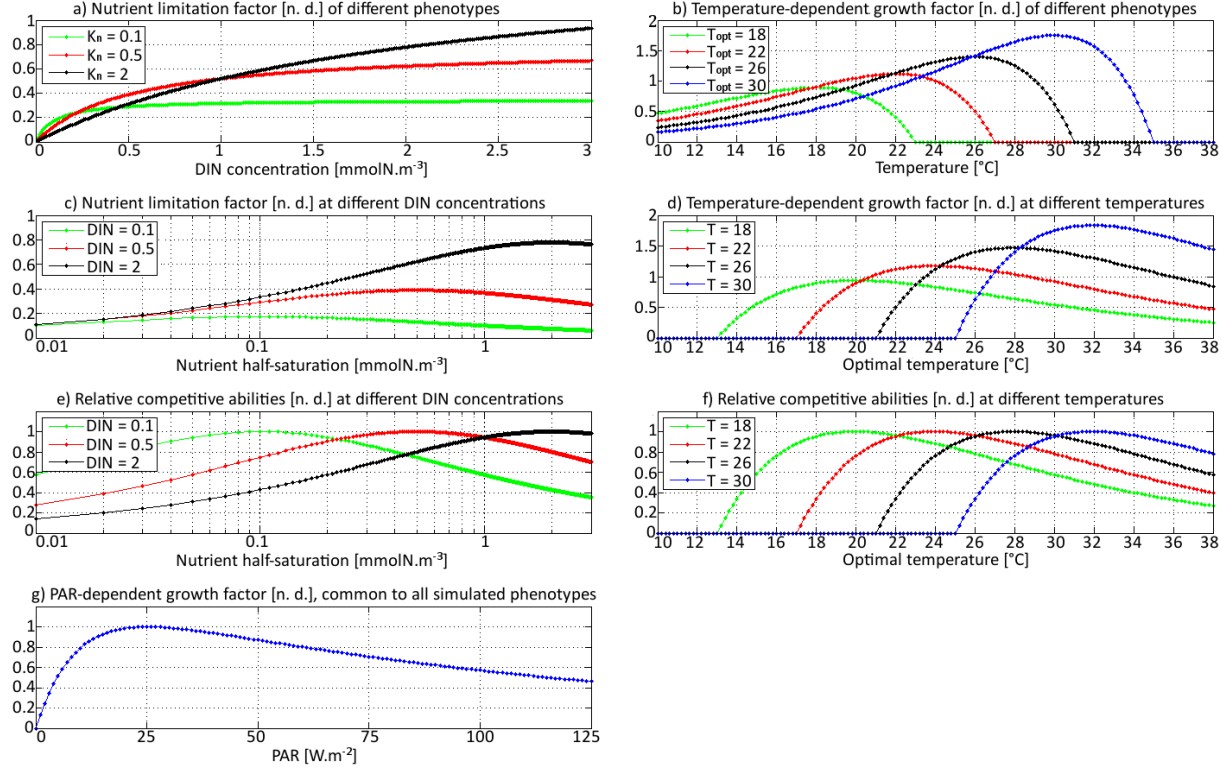

**Figure 2.** Phytoplankton growth factors $\gamma_n$ (nutrient-dependent), $\gamma_t$ (temperature-dependent) and $\gamma_i$ (PAR-dependent). a) and b) represent the growth factor as a function of nutrient concentration and temperature respectively, for different phenotypes. c) and d) represent the growth factor as a function of the corresponding trait for different values of the environmental parameter. The maximum of each curve corresponds to the phenotype most adapted to a given environment. e) and f) are normalized versions of c) and d), respectively, so that their maximum is always 1. g) is the PAR-dependent growth factor, which is common to all phenotypes in this version of SPEAD.

5   a gleaner-opportunist trade-off. The half-saturation constant $K_n = \frac{u_p^\infty}{f_p}$ (in mmolN m$^{-3}$) is the DIN concentration at which nitrogen uptake rate is equal to one half of the maximum uptake rate for the same temperature and solar irradiance. The half-saturation constant is assumed to be independent of the temperature and irradiance of the environment, as well as of the phytoplankton optimal temperature. Half-saturation makes the analysis of our results more straightforward, because it has units of concentration and can therefore be compared directly to the ambient DIN concentration. Because the concentrations

10  are always positive and span several orders of magnitude, we use a natural logarithmic scale and define $x = log\left(\frac{K_n}{K_n^0}\right)$ as our first trait axis, with $K_n^0 = 1$ mmolN m$^{-3}$ as a reference value for $K_n$. Thus, the nutrient limitation factor is:

$$\gamma_n(N,x) = \gamma_n^\infty(x)\frac{N}{K_n(x)+N} = \gamma_n^\infty(x)\frac{N}{e^x+N} \tag{15}$$

$$\gamma_n^\infty(x) = \left(\frac{K_n}{K_n^0}\right)^{1/2} = e^{x/2} \tag{16}$$





For any nutrient concentration $N$, we note that the phenotype corresponding to the largest growth rates is $x = log(N)$. This is

why, under the assumption of the gleaner-opportunist trade-off defined above, the $K_n$ defines the optimal nutrient concentration of each $x$ phenotype where they are competitively superior (Vallina et al., 2017). This result, however, is dependent on the specific model assumption that $f_p u_p^\infty$ is a constant.

The second phytoplankton trait that is allowed to mutate in SPEAD is the optimal temperature. Temperature affects microbes in two ways. One is generic and applies to the whole plankton community. An increase in temperature increases the speed of

both primary production and heterotrophic processes for thermodynamic reasons. This effect is often assumed to be exponential. In our model, the exponential factor for autotrophic primary production is $\alpha_a$ = 0.056 °C$^{-1}$, which corresponds to a Q10 of 1.75, slightly lower than the classical value of 1.88 from Eppley (1972) but higher than the values based on the metabolic theory of ecology for photosynthesis (López-Urrutia et al., 2006). The second effect of temperature is phenotype-specific. Each phenotype has an optimal temperature for growth, which is the second trait axis and is denoted by $y$. The effect of temperature

on a given phenotype (x,y) is asymmetric: at temperatures more than 5°C above $y$ growth ceases but temperatures below $y$ merely slows growth. We defined our temperature multiplicative growth factor to be as close as possible to the species-specific curves of Eppley (1972):

$$\gamma_T(T,y) = e^{\frac{(T-y)}{\Delta T}} \left( \frac{y + \Delta T - T}{\Delta T} \right) e^{\alpha_a(y - T_0)} \tag{17}$$

The temperature tolerance $\Delta T$ is set to 5°C. $T_0$ is a reference temperature with no ecological meaning. For a fixed value of

$y$, $\gamma_T(T,y)$ has a maximum at $T = y$ with a value of $e^{\alpha_a(y - T_0)}$. At $T = y + \Delta T$ and warmer, growth is impossible. For a given value of the environment temperature $T$, the phenotypes with the largest growth rates have an optimal temperature $y$ around 2°C larger than $T$. This apparent mismatch, where the dominant phenotype at temperature $T$ can grow even faster at temperatures a few degrees warmer, is both coherent with other models (Beckmann et al., 2019) and observed in nature (Thomas et al., 2012; Irwin et al., 2012).

In this study, the PAR limitation factor $\gamma_i(I)$ is the same for all phenotypes. It includes an optimal PAR ($I_{opt}$) of 25 W m$^{-2}$ and photoinhibition above this level. Our value for $I_{opt}$ is in the middle of the range considered by Follows et al. (2007) and our expression for $\gamma_i(I)$ is equivalent to theirs:

$$\gamma_i(I) = \Gamma_i^0 \left( 1 - e^{-ln(1+\chi)\frac{I}{I_{opt}}} \right) e^{-\frac{ln(1+\chi)}{\chi}\frac{I}{I_{opt}}} \tag{18}$$

$$\Gamma_i^0 = \frac{\chi+1}{\chi} e^{-\frac{1}{\chi}ln\left(\frac{1}{\chi+1}\right)} \tag{19}$$

In the above equation, $\Gamma_i^0$ is a normalization factor (to ensure that $\gamma_i(I)$ cannot exceed 1) and $\chi$ is an inhibition factor. The higher the inhibition factor, the less photoinhibition there is at irradiances larger than $I_{opt}$. In this study, we use $\chi = 12$, which is the average value in Follows et al. (2007) for large phytoplankton and corresponds well to published photoinhibition curves (Platt et al., 1980; Whitelam and Codd, 1983; Walsh et al., 2001).

For comparison with data, two additional variables can be estimated from the model: primary production and chlorophyll a concentration. Primary production ($PP$) is expressed in mgC m$^{-3}$ d$^{-1}$. Our model-based estimate is calculated by multiplying

the phytoplankton concentration and the uptake rate, normalizing from nitrogen to carbon with the 106:16 Redfield molar





ratio (Redfield, 1934) and then converting from amount of substance to mass using the molar mass of carbon ($12\,\mathrm{g.mol^{-1}}$).
Chlorophyll a concentration ($Chl$, in $\mathrm{mgCHL\,m^{-3}}$) is obtained by dividing the phytoplankton concentration in mass of carbon
by a variable carbon to chlorophyll mass ratio (C:Chl). The C:Chl ratio is estimated as in Vallina et al. (2008) using a function
of depth and time developed by Lefèvre et al. (2002), with parameter values calibrated with the observations of Goericke and

Welschmeyer (1998). At the surface, C:Chl is a sinusoidal function of the day of year, varying between a maximum of 160
$\mathrm{mgC\,mgCHL^{-1}}$ at the summer solstice and a minimum of 80 $\mathrm{mgC\,mgCHL^{-1}}$ at the winter solstice. From the depth where
$I(z,t) = 25\,\mathrm{W.m^{-2}}$ to the bottom, C:Chl decreases linearly with I(z,t) down to a value of 40 $\mathrm{mgC\,mgCHL^{-1}}$ when light is
absent.

## 2.2   Aggregate and multi-phenotype models

Traits $x$ and $y$ have an infinity of possible values. In order to solve the equations numerically, the problem needs to be simpli-
fied. Two approaches are considered here. In the "multi-phenotype" or "discrete" model approach (Fig. 1b), the trait-space is
discretized and only a finite number of phenotypes, with fixed trait values, are simulated. Phenotypes with intermediate trait
values are neglected. In the "aggregate" or "continuous" model approach (Fig. 1c), the state variables are total phytoplankton
concentration, the mean trait values, the trait variances and the inter-trait covariance. In the continuous-trait model, a specific

shape of the trait distribution must be assumed *a priori* (Wirtz and Eckhardt, 1996; Bruggeman and Kooijman, 2007). In the
discrete-trait model, the trait distribution is an emergent property and thus it does not need to be assumed beforehand.

In the multi-phenotype model, only $n_x$ values of $x$ and $n_y$ values of $y$ are allowed. The phytoplankton community is divided
into $n_x \times n_y$ phenotypes. The values of both traits are explicitly bounded by $x_{min}$, $x_{max}$, $y_{min}$ and $y_{max}$. Each phenotype
is separated from its immediate neighbors by a trait interval $\Delta x = \frac{x_{max}-x_{min}}{n_x-1}$ on $x$ or a trait interval $\Delta y = \frac{y_{max}-y_{min}}{n_y-1}$ on $y$.

Mutation fluxes at the boundaries (i.e. mutations of the phenotypes with the highest or lowest trait values leading out of the
domain) are set to zero. In the interior of our trait domain, the concentration of the phenotype with the j$^{th}$ value of $x$ and the
k$^{th}$ value of $y$, noted $P_{jk}$, is controlled by the following equation, where $a_{j,k} = u_{j,k} - g_{j,k} - m_{j,k}$ is the net growth rate:

$$\frac{dP_{j,k}}{dt} = a_{j,k}(N,T,I)P_{j,k} + \frac{\nu_x}{(\Delta x)^2}\left(2P_{j,k}u_{j,k} - P_{j-1,k}u_{j-1,k} - P_{j+1,k}u_{j+1,k}\right)$$
$$+ \frac{\nu_y}{(\Delta y)^2}\left(2P_{j,k}u_{j,k} - P_{j,k-1}u_{j,k-1} - P_{j,k+1}u_{j,k+1}\right) \tag{20}$$

In all our discrete simulations, we impose $x_{min} = -2.5$ ($K_n = 0.082\,\mathrm{mmolN\,m^{-3}}$), $x_{max} = +1.5$ ($K_n = 4.48\,\mathrm{mmolN\,m^{-3}}$),
$y_{min} = 18^\circ\mathrm{C}$ and $y_{max} = 30^\circ\mathrm{C}$. All model values of temperature and DIN concentrations are within these boundaries. We
set $n_x = 25$ and $n_y = 25$ in order to ensure that in most cases $\Delta x$ and $\Delta y$ are less than the standard deviations of $x$ and $y$,
respectively. Thus, the total number of discrete phenotypes $(x,y)$ is $25 \times 25 = 625$.

In the aggregate model, the trait distribution is assumed to be continuous. In this case, and contrary to the multi-phenotype
case, the trait axes are formally unbounded, although phenotypes with extreme trait values always have low net growth rates,
making them extremely rare. The prognostic variables are six statistical moments of the trait distribution: the total phytoplank-

5  ton concentration $P(t)$, the mean trait values $\overline{x}(t)$ and $\overline{y}(t)$, the trait variances $V_x(t)$ and $V_y(t)$ and the inter-trait covariance



$C_{xy}(t)$. They are defined as follows:

$$P(t) = \int\int p(x,y,t) \cdot dxdy \tag{21}$$

$$\overline{x}(t) = \frac{1}{P(t)} \int\int x \cdot p(x,y,t) \cdot dxdy \tag{22}$$

$$\overline{y}(t) = \frac{1}{P(t)} \int\int y \cdot p(x,y,t) \cdot dxdy \tag{23}$$

$$V_x(t) = \frac{1}{P(t)} \int\int (x - \overline{x}(t))^2 \, p(x,y,t) \cdot dxdy \tag{24}$$

$$V_y(t) = \frac{1}{P(t)} \int\int (y - \overline{y}(t))^2 \, p(x,y,t) \cdot dxdy \tag{25}$$

$$C_{xy}(t) = \frac{1}{P(t)} \int\int (x - \overline{x}(t))\,(y - \overline{y}(t))\, p(x,y,t) \cdot dxdy \tag{26}$$

The second order moments ($V_x$, $V_y$ and $C$) are difficult to interpret directly due to their dimensions. In the analyses, we thus transform variances into standard deviations ($\sigma_x$ and $\sigma_y$) and covariance into correlation ($R_{xy}$) as follows:

$$\sigma_x(t) = \sqrt{V_x(t)} \tag{27}$$

$$\sigma_y(t) = \sqrt{V_y(t)} \tag{28}$$

$$R_{xy}(t) = \frac{C_{xy}(t)}{\sigma_x(t)\sigma_y(t)} \tag{29}$$

These three diagnostic variables, along with $P$, $\overline{x}$ and $\overline{y}$, are also computed for the multi-phenotype model for comparison. The standard deviations have the same dimensions as the mean traits and can thus be compared to them. Ecologically, they represent trait diversity. Inter-trait correlation is a dimensionless number between -1 and +1, which is easier to interpret than the covariance. A correlation of -1 means above-average values of $x$ always coincide with below-average values of $y$ and vice-versa. A correlation of +1 means above-average values of $x$ always coincide with above-average values of $y$. A correlation of 0 means all combinations are equally possible (i.e. the two traits are independent).

We follow the method developed by Norberg et al. (2001), based on Taylor expansions of the uptake and net growth rates, to derive the differential equations for the moments of the trait distribution. We assume a bivariate normal distribution of traits, which is a generalization of the 1D Gaussian function. Normal distributions are observed in nature for the logarithm of size (Cermeño and Figueiras, 2008; Quintana et al., 2008; Downing et al., 2014) and are convenient assumptions for models because they produce the simplest forms for the equations (Wirtz and Eckhardt, 1996; Merico et al., 2009). The derivation is explained in detail in Appendix B. In the absence of trait diffusion, our equations are a particular case of the general equations derived by Bruggeman (2009) for multivariate normal trait distributions. In the single trait case, they are simpler than the original equations of Merico et al. (2014) and identical to the more recent formulation of Coutinho et al. (2016). The differential





equations followed by the prognostic variables are:

$$5 \quad \frac{\partial P}{\partial t} = P\left(a + \frac{1}{2}V_x\frac{\partial^2 a}{\partial x^2} + \frac{1}{2}V_y\frac{\partial^2 a}{\partial y^2} + C_{xy}\frac{\partial^2 a}{\partial x \partial y}\right) \tag{30}$$

$$\frac{\partial \overline{x}}{\partial t} = V_x\frac{\partial a}{\partial x} + C_{xy}\frac{\partial a}{\partial y} \tag{31}$$

$$\frac{\partial \overline{y}}{\partial t} = V_y\frac{\partial a}{\partial y} + C_{xy}\frac{\partial a}{\partial x} \tag{32}$$

$$\frac{\partial V_x}{\partial t} = V_x^2\frac{\partial^2 a}{\partial x^2} + 2V_xC_{xy}\frac{\partial^2 a}{\partial x \partial y} + C_{xy}^2\frac{\partial^2 a}{\partial y^2} + 2\nu_x\left(u + \frac{1}{2}V_x\frac{\partial^2 u}{\partial x^2} + \frac{1}{2}V_y\frac{\partial^2 u}{\partial y^2} + C_{xy}\frac{\partial^2 u}{\partial x \partial y}\right) \tag{33}$$

$$\frac{\partial V_y}{\partial t} = V_y^2\frac{\partial^2 a}{\partial y^2} + 2V_yC_{xy}\frac{\partial^2 a}{\partial x \partial y} + C_{xy}^2\frac{\partial^2 a}{\partial x^2} + 2\nu_y\left(u + \frac{1}{2}V_x\frac{\partial^2 u}{\partial x^2} + \frac{1}{2}V_y\frac{\partial^2 u}{\partial y^2} + C_{xy}\frac{\partial^2 u}{\partial x \partial y}\right) \tag{34}$$

$$10 \quad \frac{\partial C_{xy}}{\partial t} = V_xC_{xy}\frac{\partial^2 a}{\partial x^2} + (V_xV_y + C_{xy}^2)\frac{\partial^2 a}{\partial x \partial y} + V_yC_{xy}\frac{\partial^2 a}{\partial y^2} \tag{35}$$

The net growth rate $a$ and its derivatives with respect to traits are in all cases taken near the mean trait values ($\overline{x}$ and $\overline{y}$) and for the values of $N$, $T$ and $I$ at time $t$. The growth rate of the whole phytoplankton community depends first on the growth rate of the most abundant phenotype ("winner" of the competition), $a(\overline{x}, \overline{y}, t)$, with correction terms for the less abundant, and generally less fit, phenotypes ("losers"). Mean traits increase when larger trait values are associated with larger net growth rates, and decrease in the opposite case. The change is faster when trait variances are high. As a consequence, the overall effect of trait diversity on primary production depends on the environmental conditions. In a stable environment, high trait variances diminish the primary production, because phenotypes with low growth rates are present. Under frequent disturbances, however, high trait variances increase the short-term adaptive capacity, allowing the community to maintain mean traits close to the optimum and thereby increasing primary production (Smith et al., 2016). We note that due to covariance, the change in each trait depends on both environmental factors ($N$ and $T$). Variances decrease due to competition when mean trait values are close to the values that maximize the net growth rate ($\frac{\partial^2 a}{\partial x^2} < 0$ and $\frac{\partial^2 a}{\partial y^2} < 0$). This is most often the case, since phenotypes that are not optimal tend to be outcompeted. Trait diversity must therefore be maintained by some other process: this is the role of trait diffusion. In these equations, trait diffusion is a source of variance but does not affect the equations for phytoplankton concentration, mean traits, nor covariance. This is coherent with the fact that mutations are symmetrical (no effect on $\overline{x}$ and $\overline{y}$) and neither create nor remove biomass (no effect on $P$). Trait diffusion does not affect covariance because mutations of the two traits are independent of each other. There is no mechanistic relationship between optimal temperature and half-saturation. Mutations can create all combinations: cold-water gleaners, warm-water gleaners, cold-water opportunists and warm-water opportunists. However, by increasing variances, trait diffusion decreases the absolute value of correlation. Only the environment can correlate the traits by favoring some combinations over others. Although correlation is defined as a local quantity, for a given depth and time, it is expected to be influenced by the spatio-temporal patterns of environment variations, because local communities always contain remnants of past communities and migrants from other locations.





## 2.3 Physical setting

SPEAD 1.0 has one spatial dimension: the vertical. A depth-resolved simulation is the minimal physical setting in the ocean
to resolve the different temperature and nutrient niches and the decisive effect of the vertical mixing on the variances and the
covariance. The model is divided into 20 vertical levels, from surface to 200 m deep, with a uniform vertical step of 10 m.

Two processes can transport matter from one vertical level to another, and thus need to be added to the differential equations.
First, PON sinks at a speed of $w = 1.2\,\mathrm{m.d}^{-1}$. Sinking is solved by a first order upwind scheme. Second, tracers are verti-
cally mixed by turbulent diffusion. Vertical turbulent diffusion (called "vertical diffusion" from now on, and unrelated to trait
diffusion) tends to homogenize the spatial distribution of each tracer. It is controlled by the vertical diffusivity parameter $\kappa_z$,
expressed in $\mathrm{m}^2\,\mathrm{s}^{-1}$. The diffusion of a tracer A is $\frac{\partial}{\partial z}\left(\kappa_z(z,t)\frac{\partial A(z,t)}{\partial z}\right)$, where z is the vertical dimension. Diffusion operates
on concentrations N, P, Z and D, but not on the phytoplankton trait distribution moments, as these are not material quantities
and thus are not conserved during mixing. For instance, the mixing of two phytoplankton communities with different mean
traits creates additional variance. However, vertical mixing conserves the sum of phytoplankton trait values ($P\overline{x}$ and $P\overline{y}$) and
the sum of squared trait values ($P(V_x - \overline{x}^2)$, $P(V_y - \overline{y}^2)$ and $P(C_{xy} - \overline{xy})$). Therefore we need to convert the trait distribution
moments to $P\overline{x}$, $P\overline{y}$, $P(V_x - \overline{x}^2)$, $P(V_y - \overline{y}^2)$ and $P(C_{xy} - \overline{xy})$ before applying vertical mixing to them, and then we con-
vert them back to their original values in the next numerical time step to compute growth and loss terms. Vertical diffusion is
represented by an implicit scheme in order to avoid numerical instability. The depth-resolved model is solved in time with a
fourth-order Runge-Kutta numerical scheme.

Three environmental forcings are necessary to run the model: temperature, PAR and vertical diffusivity. All three depend
on depth and time and have been set to values from the Sargasso Sea. The forcings are seasonal. Interannual variations and
the day/night cycle are neglected. Temperature and surface PAR ($I_0(t)$) directly affect the rates of plankton growth and death.
They are set for each day using observations collected during the Bermuda Atlantic Time-series Study (BATS) (Steinberg
et al., 2001). PAR availability is assumed to decrease exponentially with depth ($I(z,t) = I_0(t)e^{-k_w z}$), with a PAR vertical
attenuation coefficient ($k_w$) of 0.04 $\mathrm{m}^{-1}$. Self-shading by phytoplankton is neglected. The vertical diffusivity $\kappa_z$ is the third
forcing. Contrary to temperature and PAR, it has not directly been observed. Therefore, the turbulent diffusivity comes from
the physical model GOTM for the Sargasso Sea (Bruggeman and Bolding, 2014). All three forcings as functions of time and
depth are shown in Fig. 3.

## 2.4 List of simulations

The simulation of the aggregate 2-trait model with mutation rates $\nu_x = 0.001$ and $\nu_y = 0.01\,°\mathrm{C}^2$ is our standard simulation for
this study. $\nu_x$ is expressed without unit because the trait axis $x$ is in logarithmic scale, but like $\nu_y$ it is a variance increase per
generation. Most of the results presented in Figures 4, 5, 6, 7 and 9 come from this standard simulation. The bulk properties
of SPEAD 1.0 (total primary production, total phytoplankton biomass, nutrient concentration) are validated using observations
from the BATS station in the Sargasso Sea (Steinberg et al., 2001; Vallina et al., 2008; Vallina, 2008). The multi-phenotype
discrete version of SPEAD is used to validate i) the assumption made in the aggregate continuous model that traits are normally



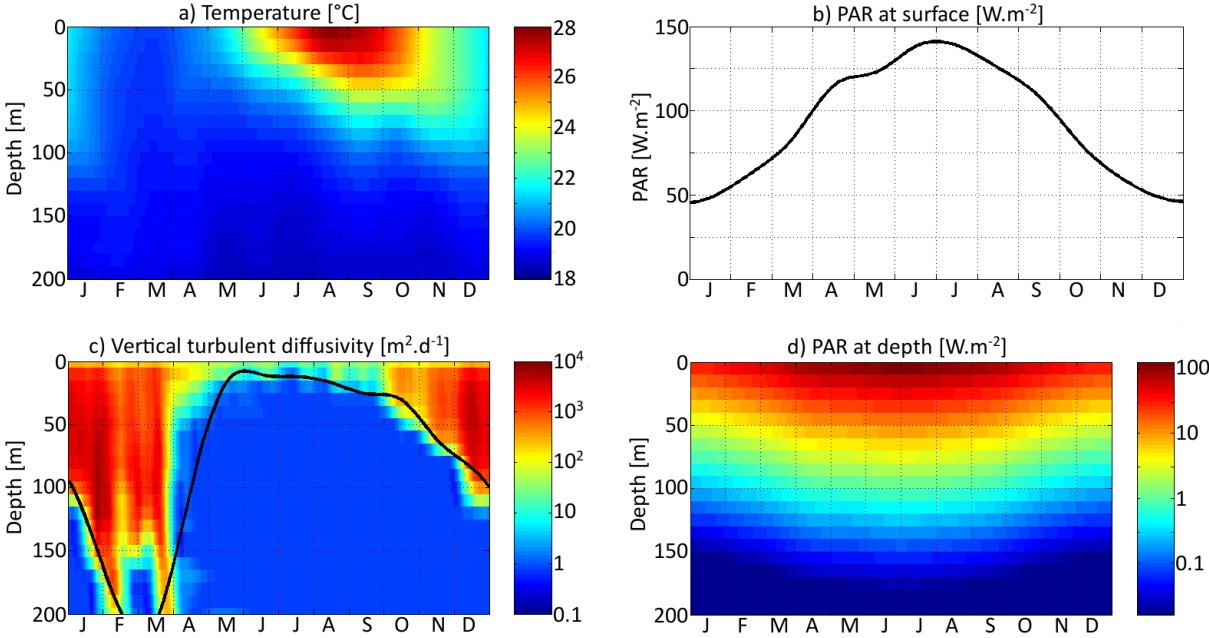

**Figure 3.** Distribution in depth and time of three environmental variables: a) Temperature, b) and d) Photosynthetically Available Radiation (PAR) and c) Vertical turbulent diffusivity. The black curve in d) represents the lower limit of the mixed layer.

distributed and ii) the simulated values and tendencies of the moments of the continuous-trait distribution. In order to better understand the behavior of the model, the standard simulation is also compared to simulations with different mutation rates

and to simulations with adaptive dynamics for only 1 trait, keeping the other trait unable to mutate but at its optimal value.

Trait diffusion is a relatively recent concept and the values of the mutation rates are not yet well calibrated by observations. To obtain a qualitative idea of the ecosystem model behavior, we tried a wide range of values for $\nu_x$ (from 0.00001 to 0.1). The largest value was chosen for its similarity to the trait diffusivity parameter chosen by Merico et al. (2014) and Chen and Smith (2018) to account for the observed trait variance. However, $\nu_x = 0.1$ allows the phytoplankton to reach a variance of

10 $V_x = 4$ in only 20 generations, since $2\nu_x$ is added to the phytoplankton population variance at each generation. This variance is the maximum allowed in the discrete model and corresponds to having half the community at each extreme of the trait axis ($x = -2.5$ and $x = +1.5$). However, laboratory experiments based on single clones show significant evolution only on timescales of hundreds to thousands of generations (Schlüter et al., 2016). For this reason we also conducted simulations with mutation rates as low as 0.00001, and a control simulation without trait diffusion at all ($\nu_x = 0$). As the mutation rate has the

15 dimension of trait squared and as the range of temperature is around three times larger than the range of nutrient concentration logarithms, we fixed the same ratio of mutation rates, $\frac{\nu_y}{\nu_x} = 10\ °C^2$, for all simulations. We checked that departing from this ratio did not qualitatively affect our results. In total, we conducted simulations for 10 sets of mutation rates, including the control case.





A 2-trait model is not simply the superposition of two 1-trait models, for at least two reasons. First, when two environmental
factors limit biomass growth, but only one is included in the model, the simulation is likely to overestimate the phytoplankton
growth rate. Second, when there is a strong inter-trait correlation, each environmental factor impacts both traits. For instance, if
the ambient DIN concentration ($N$) is below the (geometric) mean half-saturation constant ($e^{\overline{x}}$), the competition for nutrients
will select for phenotypes with a lower half-saturation constant. If at the same time the half-saturation is negatively correlated
with optimal temperature (i.e. if phenotypes with low half-saturation constants tend to also have a high optimal temperatures),
the competition for nutrients will also increase the amount of phenotypes with high optimal temperatures, in addition to the
effect of environment temperature. In the conceptual model of Savage et al. (2007), the inter-trait correlation in a 2-trait model
led to higher variances and to a considerable improvement in the ability of the mean phytoplankton traits to track optimal
values controlled by environmental conditions compared with 1-trait models. In order to know whether these results also apply
to our model, we compare the dynamics of traits $x$ and $y$ in SPEAD to the dynamics of simplified 1-trait models where either
$x$ or $y$ vary between phenotypes and the other trait is optimized instantaneously (i.e. set to the optimal value at each location
and time given the environmental conditions).

The time step for our simulations is 6 hours. At the first time step and at all vertical levels, DIN concentration is initial-
ized to 1.8 mmolN m$^{-3}$, phytoplankton and zooplankton concentrations to 0.1 mmolN m$^{-3}$, and PON concentration to 0.0
mmolN m$^{-3}$. The total amount of nitrogen in the water column is conserved, and every loss below 200 m due to PON sinking
is compensated by an equivalent gain of DIN, also at 200 m. Mean logarithm of half-saturation and optimal temperature are
initialized at -0.5 (corresponding to $K_n = 0.61$ mmolN m$^{-3}$) and 24°C respectively, with initial standard deviations of 0.1 and
0.3°C. Each simulation is run for at least 3 years and until convergence is reached. Our convergence criterion is that, for every
day of year and every depth level, the difference between the two last years should be less than 0.1% for P, $V_x$ and $V_y$, less than
0.1% of the modeled range for $\overline{x}$ and $\overline{y}$ and less than 0.001 for $R_{xy}$. In other words, convergence is achieved when the seasonal
cycle of the model state variables is repeated from year to year. The results shown are in all cases from the last simulated year.
We checked that total nitrogen was the only feature in the initial conditions that affected the results.

## 3 Results

### 3.1 Bulk modeled properties and comparison with observations

The first step to validate the SPEAD model is to compare some bulk properties with observations from the Sargasso Sea.
In Fig. 4, the primary production, chlorophyll, DIN and PON concentrations of the aggregate model are compared to a 10-
year climatology of monthly observations of primary production, chlorophyll, nitrate and PON concentrations (Vallina et al.,
2008). With carefully chosen values for the parameters of the ecosystem model (Table 2), the model and observations agree
well, although some minor discrepancies exist, due to the simplicity of our parameterizations.

Primary production is the state variable best reproduced by the model, with a maximum around 10 mgC m$^{-3}$ d$^{-1}$ in the first
50 m in February and March in both the model and the observations. From May to September, primary production spreads
slightly more in depth but is overall around half its maximal value. Primary production is negligible deeper then 80 m.



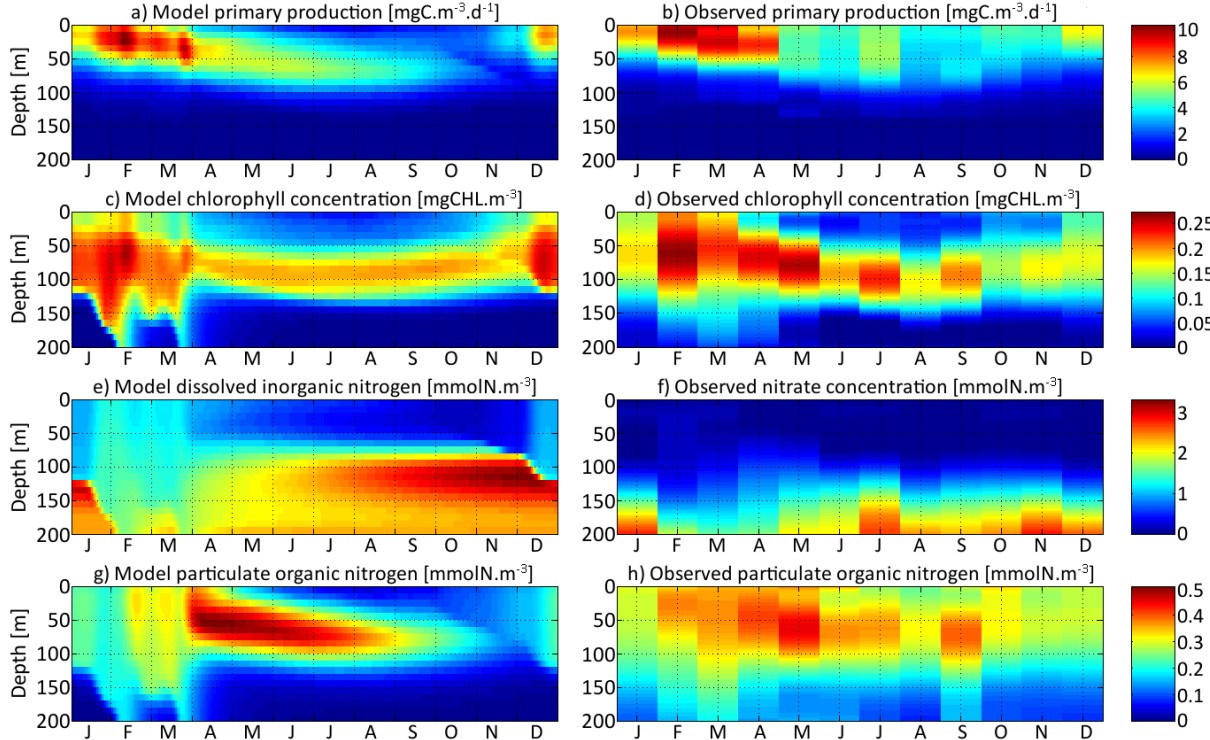

**Figure 4.** Distribution in depth and time of a) model primary production, c) chlorophyll concentration, e) dissolved inorganic nitrogen concentration and g) particulate organic nitrogen concentration. Each variable is compared with equivalent observations in the Sargasso Sea (b, d, f and h).

Chlorophyll concentration is reproduced with the right order of magnitude, an absolute maximum correctly located in February between 50 m and 80 m deep, at about 0.25 mgCHL.m$^{-3}$, and a deep chlorophyll maximum around 100 m in Summer. However, the model chlorophyll concentration is lower than the observations in Spring and higher in late Autumn. Chloro-

phyll concentration begins to increase in December in the model but only in February in the observations. Both chlorophyll concentration and primary production are proportional to phytoplankton concentration. The reason for the temporal mismatch between SPEAD and the observations in chlorophyll concentration, but not in primary production, must then be related to the temporal variability of the other factors affecting these two quantities: the nitrogen uptake rate and the C:CHL ratio. The relatively high primary production and very low chlorophyll concentration observed in December might be accounted for better if

the uptake rate were faster in December than in February, despite the lower availability of nutrients and light, or if turbulence were included in the estimation of the C:CHL ratio so that it reaches its lowest value in February, when the waters are best mixed and phytoplankton cannot stay close to the surface  (Taylor et al., 1997), rather than in December, when the surface light intensity is minimum (Lefèvre et al., 2002; Jakobsen and Markager, 2016).





Dissolved Inorganic Nitrogen is compared to the observed nitrate concentration, knowing that this form of DIN dominates at
depth but co-occurs with nitrite and ammonium, which are also components of DIN. The modeled DIN and the observed nitrate
concentrations share the same range, with a maximum of 2.8 mmol m$^{-3}$ in the observations and 3.3 mmol m$^{-3}$ in the model.
Another common point between the model and the observations is that both reach a maximum at the bottom of our setting,
at 200 m, with concentrations between 2.1 and 2.8 mmol m$^{-3}$ during most of the year. Because of the strong vertical mixing,
from February to April, the concentrations are lower, between 1.5 and 2.0 mmol m$^{-3}$, but still a maximum. However, from
June to January, the modeled DIN concentration exhibits a second maximum that is absent from the observations. This second
maximum is located just below the euphotic layer, between 100 m and 150 m deep, with values as high as 3.3 mmol m$^{-3}$
in late November. Both modeled and observed concentrations are minimal at the surface, due to the nitrogen uptake by
phytoplankton. However, their values diverge by more than one order of magnitude. Modeled DIN concentrations at the surface
vary between 0.18 and 1.34 mmol m$^{-3}$, with a mean of 0.67 mmol m$^{-3}$, whereas observed nitrate concentrations vary between
0 and 0.11 mmol m$^{-3}$, with a mean of 0.03 mmol m$^{-3}$. We assume that these discrepancies are due to the contribution of
ammonium, and possibly nitrite, since the few studies reporting measured concentrations of ammonium in the Sargasso Sea
(Menzel and Spaeth, 1962; Brzezinski, 1988) showed that ammonium was more homogeneously distributed in the upper 200
m than nitrate and was the dominant form of dissolved nitrogen from surface to 100 m deep.

Particulate organic nitrogen distributions from the model and observations are relatively similar, with a maximum around 0.5
mmol m$^{-3}$ (0.45 mmol m$^{-3}$ in the observations, 0.51 mmol m$^{-3}$ in the model) in April and May at depths between 30 m and
80 m. However, the seasonality and vertical gradients are much larger in the model, where particles are very rare in Autumn
and nearly absent at depths greater than 100 m, whereas observed PON concentrations are never below 0.08 mmol m$^{-3}$. The
observations might be better explained if a minority of particle production went to a slowly remineralizing refractory pool,
enabling them to stay during the whole year and to reach greater depths  (Aumont et al., 2017), but we did not increase the
complexity of the particle parameterization because this is not the focus of our study.

### 3.2   Trait distribution of SPEAD and comparison with a multi-phenotype model

The second step to validate the aggregate SPEAD model and the only validation of its bivariate trait distribution is done by
5   comparing it to the multi-phenotype model. Although both are models and thus simplify reality in similar ways, the multi-
phenotype model is used as a reference for two reasons. First, it is more intuitive than the aggregate model, with birth and
death processes and mutations to the nearest neighbors as the only terms in the equations. Therefore, the moments of the trait
distribution in the multi-phenotype model can be used as a control to confirm that the equations of the aggregate model are
correct. Second, the multi-phenotype model does not assume any particular trait distribution shape and can be used to validate
10   the *a priori* assumption of the aggregate model that the trait distribution is a bivariate normal distribution.

The spatial and temporal patterns of phytoplankton concentration, mean traits, trait standard deviations and inter-trait cor-
relation for the standard simulation with mutations rates of $\nu_x = 0.001$ and $\nu_y = 0.01$ °C$^2$ are shown in Fig. 5. The value of $\overline{x}$
varies between -0.83 ($K_n = 0.44$ mmolN m$^{-3}$) and +0.6 ($K_n = 1.82$ mmolN m$^{-3}$), with standard deviations between 0.31 and
0.77. The value of $\overline{y}$ varies between 22.0 °C and 26.1 °C, with standard deviations between 0.81 °C and 1.92 °C. By compari-





son, the modeled DIN concentration varies between 0.18 and 3.31 mmolN m$^{-3}$, and the water temperature varies between 18.5 and 27.8 °C. As expected, the mean trait values remain consistently within the range of the environmental drivers to which they adapt. Because the best competitor at a given time and depth needs tens of generations to become dominant after having been

a rare phenotype, the mean traits react with a delay of 1 to 2 months and with a lower amplitude than their drivers. Cold-water opportunists (high $x$-trait, low $y$-trait) dominate in Winter and Spring throughout the water column. In Summer, they are slowly replaced by warm-water gleaners (low $x$-trait, high $y$-trait) in the upper 70 m but retain dominance at greater depths, where their half-saturation constants continue increasing and their optimal temperatures continue decreasing. The coexistence of two very distinct communities in Summer and early Autumn is made possible by the intense stratification, which creates a physical

barrier between the different depth levels. In late Autumn, the two communities are rapidly mixed by the vertical turbulent diffusion, producing a peak in the standard deviation of each trait, in other words a peak in the local (alpha) diversity. Then, as the water column becomes more homogeneous, competition selects for a single dominant phenotype, reducing the trait diversity until the next Autumn. Inter-trait correlation is negative at all times and depths, due to the negative correlation of the environmental drivers. High DIN concentrations generally coincide with low temperatures, favoring cold-water opportunists.

This happens during Winter because turbulent mixing brings nutrient-rich cold waters from the deep layers up to the surface. During Summer, the consumption of nutrients by primary producers leads to a coincidence of warm temperatures with low DIN concentrations at the surface. In late Autumn, the negative correlation reaches its maximum absolute value when the two main communities are suddenly mixed. During the rest of the year, trait diffusion progressively reduces the inter-trait correlation.

In Fig. 6, the state variables of the aggregate model are compared to the trait distribution moments of the discrete model.

The discrete model is considered as a "truth" and the difference between the two models as an "error", positive if the value is higher in the aggregate model. The aggregate model reproduces $P$ and $\bar{x}$ very precisely, with linear determination coefficients (R$^2$) of 0.998 and 0.988 respectively. The biases (mean error) are very low: +0.0005 mmolN m$^{-3}$ on $P$ and -0.04 on $\bar{x}$. The bias of $\bar{y}$ is larger, at -0.50 °C, but the coefficient of determination is still very high, at R$^2$ = 0.862. The error is largest in the deep community in Summer and early Autumn, reaching a maximum of -1.51 °C in early September around 100 m. The

most likely reason why the aggregate model underestimates $\bar{y}$, but not $\bar{x}$, is that the response of phytoplankton to temperature is asymmetrical. Increases in the environment temperature put more selective pressure on the phytoplankton community than decreases. This feature is poorly taken into account by the aggregate SPEAD model because of its assumption that traits are normally distributed. There is a mismatch between the asymmetrical shape of temperature niches and the imposed symmetrical shape of the distribution of optimal temperatures ($y$-trait) in the aggregate model. This mismatch does not happen for the $x$-

trait. As is typically the case in aggregate models, there are more errors in the higher order moments, in our case the standard deviations and the inter-trait correlation. The coefficients of determination for $\sigma_x$, $\sigma_y$ and $R_{xy}$ are R$^2$ = 0.813, R$^2$ = 0.462 and R$^2$ = 0.896 respectively. Their biases are -0.04, +0.09 °C and -0.001 respectively, which is around 10% of the mean value for $\sigma_y$ and $\sigma_x$ and negligible for $R_{xy}$. All three variables decrease much faster in early Winter in the aggregate model than in the multi-phenotype model. Additionally, in Summer, there is a strong discrepancy for $\sigma_y$. In the discrete model, $\sigma_y$ can reach as low as 0.57 °C but in the aggregate model it is never less than 0.81 °C and very rarely less than 1.0 °C.



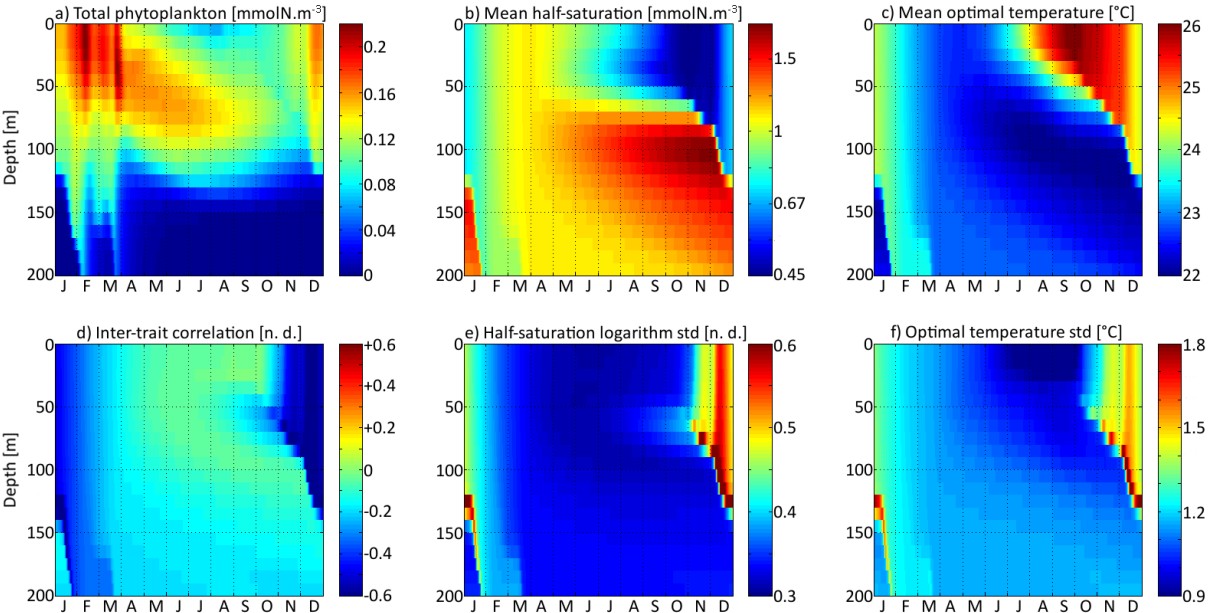

**Figure 5.** Distribution in depth and time of trait distribution moments for $\nu_x = 0.001$: a) phytoplankton concentration, b) (geometric) mean half-saturation, c) optimal temperature, d) inter-trait correlation, e) half-saturation logarithm standard deviation and f) optimal temperature standard deviation. For readability, the mean value of trait $x$ is transformed into a nitrogen concentration in b). To speak properly and contrary to other means present in this study, the "mean half-saturation" is a geometric mean, not an arithmetic mean.

The main errors on $\sigma_x$, $\sigma_y$ and $R_{xy}$ are caused the aggregate model's assumption of a multi-variate normal distribution, which is not strictly correct based on the results of the discretely resolved model. In Fig. 7, we show in a 2D color plot how the two traits are distributed in the discrete model at three different depths (surface, 50 m and 100 m) at the end of

5 each season (March, June, September, December). This distribution is compared with the bivariate normal distribution of the aggregate model, represented by ellipses. In March, when the waters are well-mixed, and in June, when the stratification has just begun, the traits are normally distributed and the two models agree. There is only a small error on the distribution of optimal temperature. In March at all depths and in June at 100 m, the normal distribution of the aggregate model contains more phenotypes with low optimal temperatures than the distribution of the discrete model. In Summer, the traits are also

normally distributed near the surface. However, the distribution of optimal temperature is markedly right-skewed deeper in the water column. Optimal temperatures below that of the most common phenotypes are extremely rare whereas those larger than this level are more common. The aggregate model has its $\overline{y}$ below the peak of the multi-phenotype model and has a much larger $\sigma_y$. This is the largest error for both a mean trait value and a trait standard deviation in this study. Since nothing similar occurs with half-saturation, this error must be linked with the right skew in the temperature-dependent growth factor

when expressed as a function of optimal temperature (Fig. 2d). In stable environments with little change in temperature with time and little vertical mixing, the distribution of optimal temperature tends to become naturally right-skewed. However, our





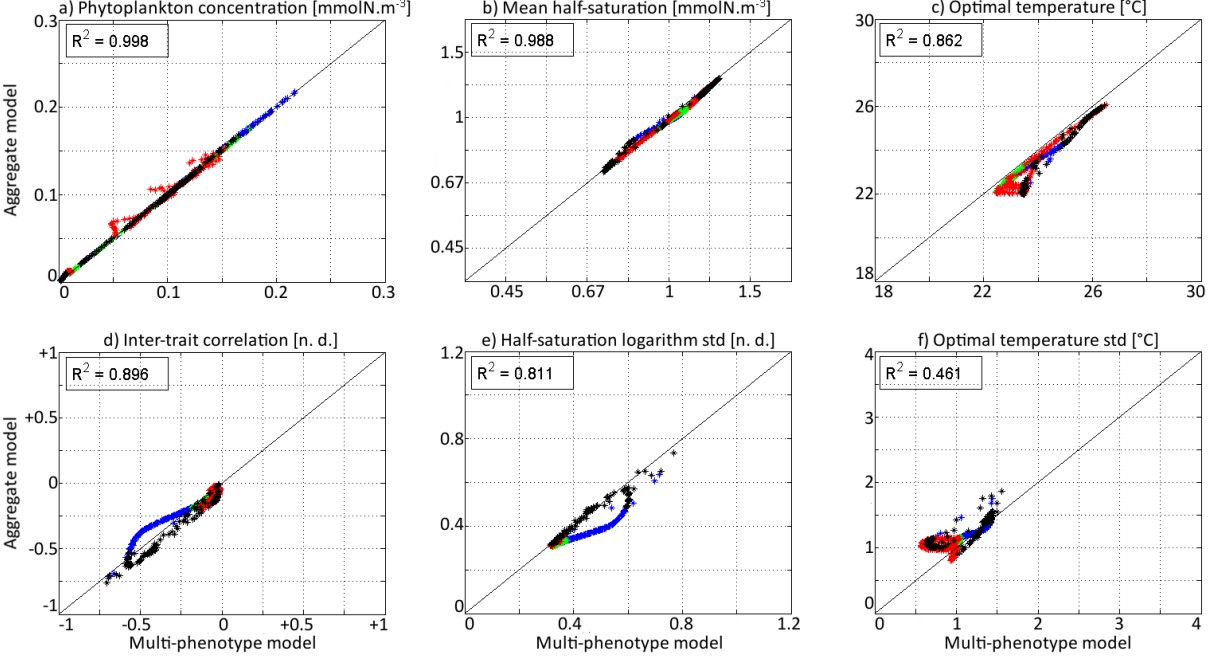

**Figure 6.** Comparison at all depths and time of the aggregate and multi-phenotype model state variables for $\nu_x = 0.001$: a) total phytoplankton concentration, b) (geometric) mean half-saturation, c) mean optimal temperature, d) inter-trait correlation, e) half-saturation logarithm standard deviation and f) optimal temperature standard deviation. Blue is Winter, green is Spring, red is Summer and black is Autumn.

results show that re-mixing (in late Autumn), fast environmental change (near the surface) and trait diffusion can reduce or eliminate this skew, so that the trait distribution is often close to normality. In December during the re-mixing phase, the trait distribution completely deviates from normality and becomes bimodal, with a community of warm-water gleaners and a

community of cold-water opportunists co-occurring throughout the water column. At this time the standard deviations and the inter-trait correlation are at their annual maxima. The moments of the trait distribution at that time are very well captured by the aggregate model. However, assuming a normal trait distribution is not only wrong in terms of ecological description but also leads to incorrect dynamics during Winter. In Winter, the ecological selection in the now mixed waters reduces trait diversity, and trait diffusion reduces the inter-trait correlation. These processes occur faster in the aggregate model than in the multi-

phenotype model (see Fig. 6) because the selective pressure is larger for a normal distribution than for a bimodal one. From a mathematical point of view, this can be shown in a simplified 1-trait model. Selection through competition reduces the trait variance at a speed equal to $\frac{1}{2}(M_4 - V_x^2)\frac{\partial^2 a}{\partial x^2}$ (this is a 1-trait unskewed version of equation B7), where $M_4$ is the fourth order moment or "kurtosis". In a Gaussian distribution, $M_4 = 3V_x^2$. Bimodal distributions have a lower kurtosis, therefore they are affected more slowly by ecological selection. By design, the aggregate model cannot account for this effect because it assumes

a unimodal Gaussian distribution. From a more ecological point of view, it can be noted that in order to replace a bimodal distribution by a unimodal one with a smaller variance, a previously rare intermediate phenotype must rise to prominence





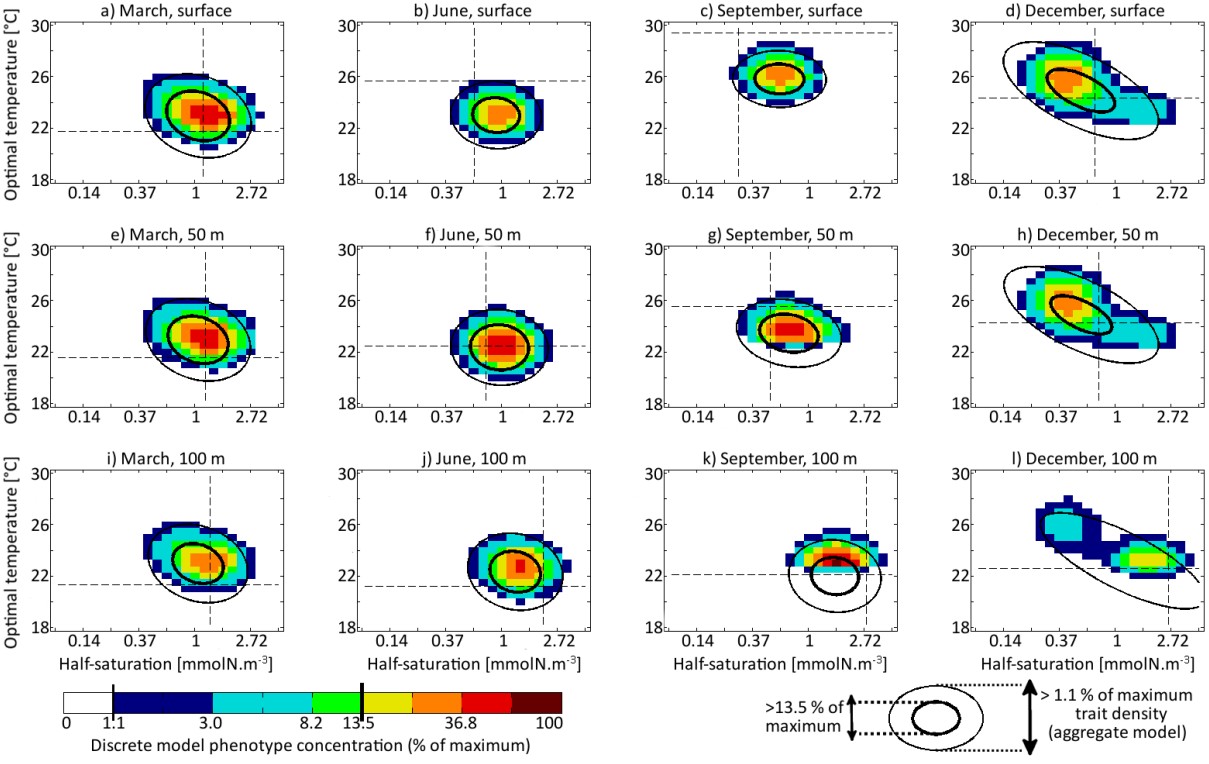

**Figure 7.** Concentrations of each phenotype of the multi-phenotype model (color) compared with lines of equal density of the aggregate model. Subplots correspond to days 71, 161, 251 and 341, and depths of 0, 50 and 100 m. Dashed lines indicate the optimal competitor.

and previously dominant phenotypes must become rare, which is a dramatic change. By comparison, in an already unimodal Gaussian distribution, reducing the variance only means making rare and extreme phenotypes even rarer.

### 3.3 Trait dynamics with different mutation rates

In this section, we compare the results of simulations conducted with 9 different sets of mutation rates, from $\nu_x = 0.00001$ and $\nu_y = 0.0001\ °C^2$ to $\nu_x = 0.1$ and $\nu_y = 1.0\ °C^2$. A control simulation with no trait diffusion is also included. That amounts to a total of 10 simulations. The ratio of mutation rates, $\frac{\nu_y}{\nu_x} = 10\ °C^2$, is the same in all simulations. This comparison highlights the unique role played by trait diffusion in SPEAD, even at very low mutation rates.

For each simulation, Fig. 8 shows the values of depth-integrated primary production per year and the yearly averaged values and ranges of $\overline{x}$, $\overline{y}$, $\sigma_x$, $\sigma_y$ and $R_{xy}$. Additional diagnostics are presented in Table 3. The number of years to converge to a steady state, beyond being just a numerical issue, can also serve as an ecological indicator of the time needed to damp a perturbation or to adapt to a new physical setting, although, by design, our convergence times cannot be less than 3 years. The Control simulation does not fully converge even after decades and we decided to run it 38 years, which is the convergence time of the simulation with the lowest non-zero mutation rates. However, we find that the trait diversity of the Control scenario



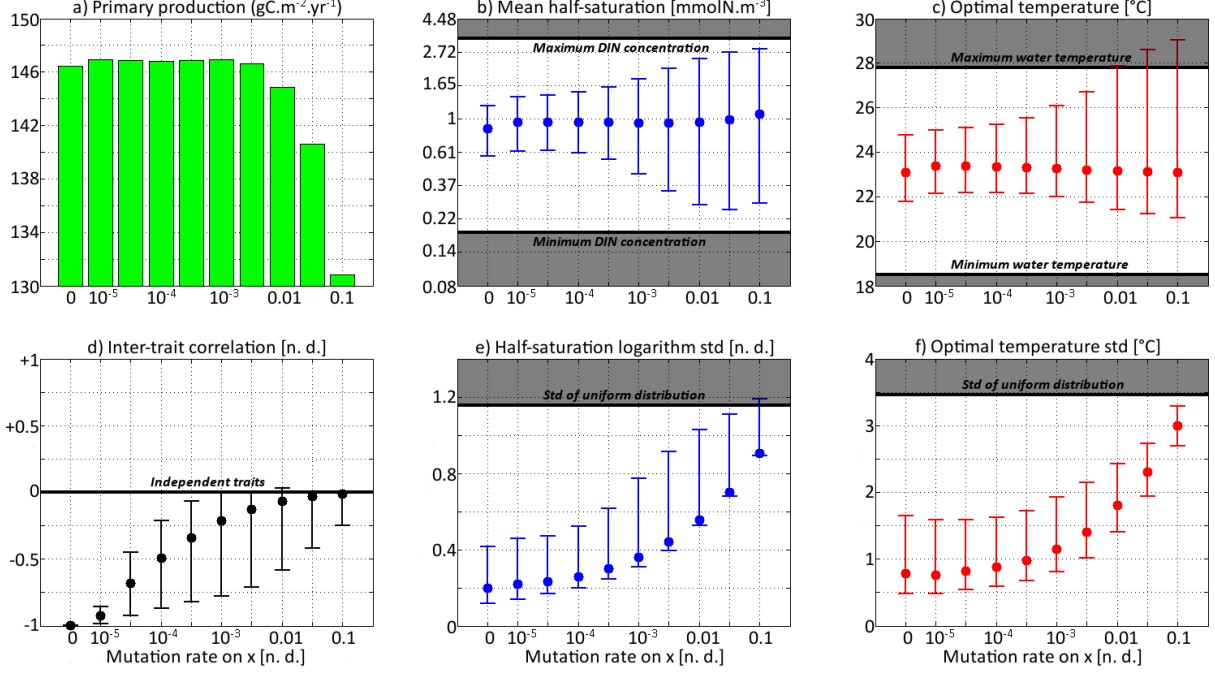

**Figure 8.** Primary production and trait distribution moments for different mutation rates. The ratio $\frac{\nu_y}{\nu_x}$ is kept constant and equal to $10°C^2$. Moment ranges are represented by error bars, and their mean by dots. The mean traits are compared to the extreme values of their environmental drivers (dissolved inorganic nitrogen concentration and temperature). The trait standard deviations are compared to their values in a uniform distribution within the boundaries of the discrete model.

does not collapse to zero as we expected because the standard deviations of its $x$ and $y$ traits ends up being higher than their initial values and continue slowly increasing. Yet, the standard deviations of the Control scenario is significantly smaller than for the scenarios with non-zero trait diffusion. For either trait, the maximum value of $\frac{V}{2\nu}$ is the number of generations required to reach the highest trait variances of the simulation in the absence of ecological selection and with trait diffusion as the only source of variance. Although highly idealized, this number is a proxy for the timescale of evolutionary processes. Table 3 also

assesses whether bimodality occurs at some point in the year in the discrete model and whether the mean traits come within one standard deviation of the discrete model boundaries.

  Primary production is around 146.9 gC m$^{-2}$ yr$^{-1}$ for all mutation rates between $\nu_x = 0.00001$ and $\nu_x = 0.001$, then decreases at higher trait diffusivities to finally reach 130.8 gC m$^{-2}$ yr$^{-1}$ for $\nu_x = 0.1$. The primary production of the control simulation is 146.5 gC m$^{-2}$ yr$^{-1}$. This result agrees with the model of Chen et al. (2019) as applied to the North Pacific.

Their phytoplankton community was characterized by one trait, cell size, which is somehow related to our half-saturation trait $x$. They found that primary production was diminished when $\nu_x$ increased, but they only considered relatively high mutation rates between 0.01 and 0.1, as well as a control simulation. Under relatively stable conditions, we find that fast mutation rates ($\nu_x > 0.01$) are a drawback for primary production because they promote large trait variances, allowing non-competitive





**Table 3.** Convergence time and various properties of SPEAD 1.0 simulations with different mutation rates

| $\nu_x$ | $\nu_y$ | Convergence time | $\max\left(\frac{V}{2\nu}\right)$ | Out of range | Bimodality | Adapts faster |
|---|---|---|---|---|---|---|
| – | [°C$^2$] | [years] | [generations] | | (discrete model) | with 2 traits |
| 0 | 0 | – | – | No | Yes | Yes |
| 0.00001 | 0.0001 | 38 | 12512 | No | Yes | Yes |
| 0.00003 | 0.0003 | 14 | 4213 | No | Yes | Yes |
| 0.0001 | 0.001 | 10 | 1363 | No | Yes | Yes |
| 0.0003 | 0.003 | 7 | 631 | No | Yes | Yes |
| 0.001 | 0.01 | 5 | 297 | No | Yes | No |
| 0.003 | 0.03 | 4 | 139 | No | Yes | No |
| 0.01 | 0.1 | 4 | 53 | Yes | No | No |
| 0.03 | 0.3 | 4 | 21 | Yes | No | No |
| 0.1 | 1 | 3 | 7 | Yes | No | No |

phenotypes (i.e. under-performers) to proliferate. However, phytoplankton mutating very fast could be invaded by phenotypes
mutating more slowly. Therefore we do not expect them to be common in nature.

In the simulations with $\nu_x = 0.01$, $\nu_x = 0.03$ and $\nu_x = 0.1$, the mean trait values remain close to their environmental drivers
and their range over the year is as wide as that of DIN concentration and temperature, respectively. On average, the community
adapts nearly instantaneously to its environment. However, the cost for this apparent success in fast-tracking the environmental
conditions is that the standard deviations of the trait distribution are very high and close to that of a uniform distribution
between the trait boundaries of the discrete model. Given that the trait domain of the discrete model is already wider than the
ranges of DIN concentration and in-situ temperature, this result suggests that either 1) phenotypes that are maladapted at all
depths and throughout the year are common (explaining the low primary production) or 2) that we have reached the limit of
validity of the aggregate approach. These simulations do not have skewed or bimodal distributions, and their correlations are
5  negligible, even in December when the stratification is broken, because trait diffusion is a symmetrical process that constantly
replenishes all rare phenotypes, including warm-water opportunists and cold-water gleaners that are maladapted at all depths
and during all the year. The simulations with large mutation rates converge in 3 or 4 years and can sustain their variances in
less than 60 generations, that is, in less than a year. They use mutations to follow the seasonal cycle of their environment faster
than the usual timescales of evolution, even for phytoplankton (Schlüter et al., 2016).
10  When the mutation rates are lower, the mean traits still vary during the year but not as much nor as fast as the physical
environment, and no phenotypes are found outside of the trait domain of the discrete model. With $\nu_x$ at 0.001 or lower, several
years are required to sustain the variance and to converge to a seasonally stable state. In this case, the mutations create variance
over the long term, facilitating the ecological successions of phenotypes seasonally and the adaptive evolution inter-annually.
However, low mutation rates do not allow the community to evolve seasonally. Bimodality is present in the discrete version,





at least during the late Autumn mixing, and lasts longer as the mutation rates decrease. The variances increase when the trait diffusivity parameters increase, which is what trait diffusion was designed for. We note that, contrary to chemostat models (Merico et al., 2009), SPEAD 1.0 does not require trait diffusion to sustain a positive trait diversity: the trait variances do not collapse to zero even in the absence of trait diffusion. The late Autumn mixing is a source of variance in its own right, avoiding

the collapse of trait diversity even in the Control simulation. However, the trait standard deviations in the Control case are very low, between 0.12 and 0.41 for $x$. Trait diversity appears even lower when accounting for the fact that correlation between $x$ and $y$ is blocked at -1. The $x$ and the $y$ traits totally determine each other, as if there were only one trait and no extra degree of freedom. The only active phenotypes are located on a straight line. Trait diffusion is not necessary to sustain variance, but it is necessary to allow the model to explore the entire trait space and to adapt to entirely new sets of environmental conditions.

Increasing trait diffusion to $\nu_x = 0.00001$ does not lead to any significant increase in trait variance, which keeps being extremely low. The variance is still overwhelmingly controlled by the December mixing, producing very large correlations. Just above this level, the cases from $\nu_x = 0.00003$ to $\nu_x = 0.0003$ share features that are coherent with our expectations on the effect of inter-generational mutations: a high primary production, a moderately high but never total correlation and timescales of a few years ( 100s to 1000s of generations) to adapt to their environment.

## 3.4  Trait dynamics compared with 1-trait models

In Figs. 9 and 10, the trait distribution moments of SPEAD at the surface are compared with the environmental drivers (DIN concentration and water temperature) and with the outputs of two single-trait aggregate models, where only the half-saturation constant or only optimal temperature is allowed to vary between phenotypes, subject to trait diffusion. Figure 9 shows the comparison for the standard values of the mutation rates: $\nu_x = 0.001$ and $\nu_y = 0.01$ °C$^2$. However, the differences between

2-trait and 1-trait models are likely to be larger when correlations between $x$ and $y$ are large. Therefore, in Fig. 10, we compare the 2-trait and 1-trait model distributions obtained with the lowest non-zero mutation rates: $\nu_x = 0.00001$ and $\nu_y = 0.0001$ °C$^2$, which lead to inter-trait correlations between -0.8 and -1 in the 2-trait model.

With standard mutation rates, the trait dynamics are very similar in all three models. The 2-trait model has slightly lower standard deviations than the 1-trait models during some parts of the year, but the difference is always within 10%. The seasonal

patterns are very similar in both timing and amplitude. The greatest differences are found in Summer, from mid-June to mid-August, when the 1-trait model with adaptive dynamics for half-saturation has a greater phytoplankton concentration by as much as 29% and a lower nutrient concentration by as much as 24% compared to the other two models, which concentrations are very similar to each other. This result means that at the onset of Summer, the most important factor decreasing the ability of phytoplankton to grow is not the lack of nutrients, but temperature itself. In other words, the phenotypes that dominated

in Spring decline, not because they are not adapted to oligotrophic conditions, but because they are not adapted to the high temperatures of Summer and the growth rate of a phenotype declines sharply when temperature exceeds its optimal value. This effect is negligible at the highest mutation rates, because in this case the community is able to evolve and adapt very quickly to the Summer warming, but becomes more important as the mutation rates, and hence the optimal temperature variances, decrease.



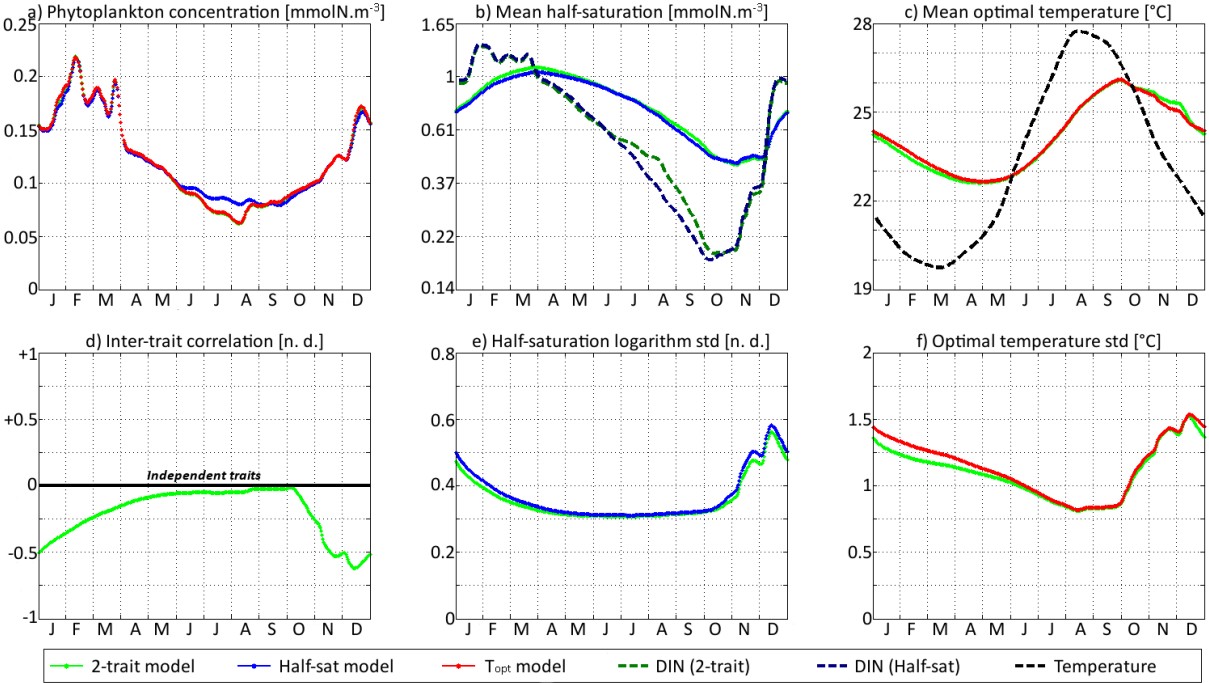

**Figure 9.** SPEAD state variables at surface for $\nu_x = 0.001$ (standard) compared with the state variables of 1-trait models: a) total phytoplankton concentration, b) (geometric) mean half-saturation, c) mean optimal temperature, d) inter-trait correlation, e) half-saturation logarithm standard deviation and f) optimal temperature standard deviation. The dashed lines represent the environmental drivers.

The differences between models are larger at low mutation rates. With the smallest non-zero mutation rates, the Summer difference in phytoplankton biomass increases. The 1-trait half-saturation model has now a phytoplankton biomass as much as 57% greater and a DIN concentration as much as 30% smaller than in the other models. Trait variances are again lower in the 2-trait model during most of the year but sometimes exceed the 1-trait variances during the Autumn mixing. However, the most

5   notable change is that the seasonal amplitude of mean half-saturation ($\overline{x}$) is now 56% higher in the 2-trait model than in the 1-trait half-saturation model. Having a second trait allows the ecosystem to adapt faster to environmental changes. This effect is even more notable when considering that both the half-saturation variance and the nutrient-mediated selective pressure are lower in the 2-trait model. This effect does not extend to the other trait, although the mean optimal temperature of the 2-trait model and that of the 1-trait optimal temperature model sometimes show slight departure from each other, in a seasonally

10  dependent way.

The effects described above are related to inter-trait correlation, which is driven by correlated environmental conditions and becomes very large in the case of low mutation rates. Equations 30 to 35 can help understand the effect of trait correlation on the seasonality of mean traits and trait variances. In the mean-trait equations (31 and 32), correlation implies that both temperature and DIN concentration drive changes in both mean trait values. The covariance term can either accelerate or slow down the response of each mean trait, but generally the sign of covariance is such that the change is accelerated. This is what occurs from



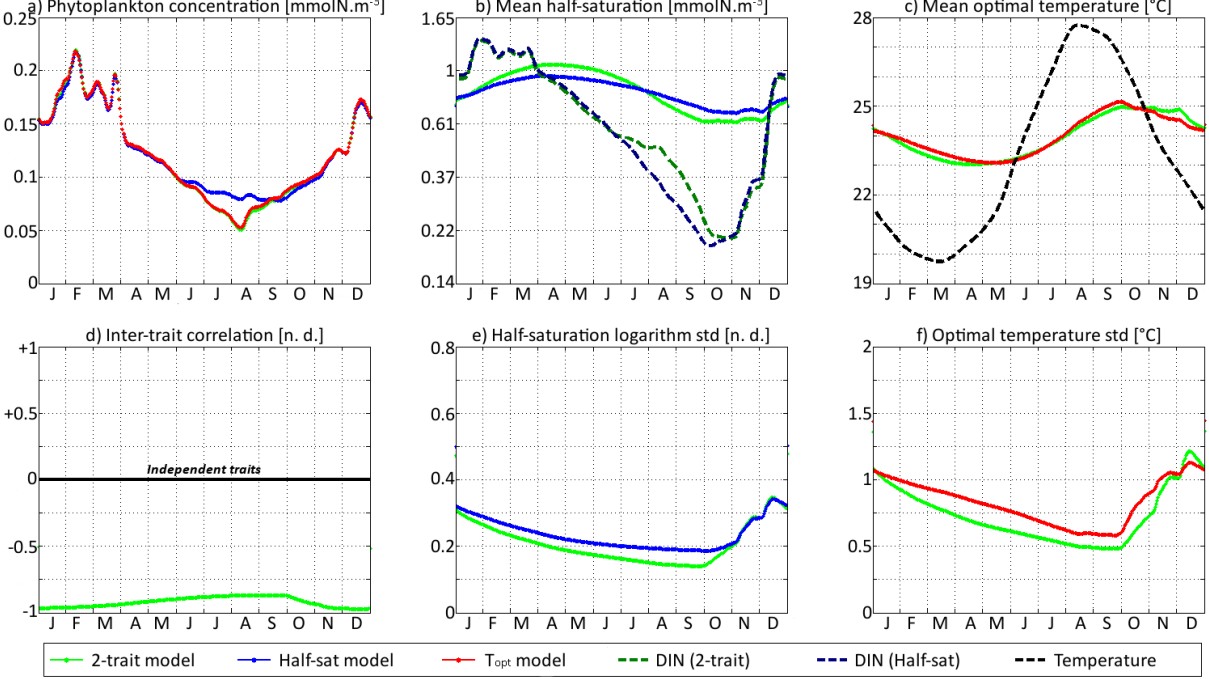

**Figure 10.** SPEAD state variables at surface for $\nu_x = 0.00001$ (low mutation rate) compared with the state variables of 1-trait models: a) total phytoplankton concentration, b) (geometric) mean half-saturation, c) mean optimal temperature, d) inter-trait correlation, e) half-saturation logarithm standard deviation and f) optimal temperature standard deviation. The dashed lines represent the environmental drivers.

December to March, when the environment selects for higher half-saturation constants and lower optimal temperatures, and the environmentally induced negative correlation between traits further accelerates this adaptation. From June to October, the same effect occurs but is significant only for half-saturation. During these months, the 2-trait model actually experiences a slower

increase in optimal temperature then the 1-trait model because it has a smaller variance and because the selective pressure of high temperatures is much sharper than the nutrient-mediated pressure conveyed by the correlative term. In November, correlation has the opposite effect: as the temperature decreases while the DIN concentration remains low, the environment at that time selects for both low optimal temperature and low half-saturation, and the negative correlation prevents optimal temperature from decreasing.

The effect of correlation on variance is even more convoluted. In Equations 33 and 34, inter-trait correlation adds a second variance-reducing competition term ($C_{xy}^2 \frac{\partial^2 a}{\partial y^2}$ and $C_{xy}^2 \frac{\partial^2 a}{\partial x^2}$, respectively, both very likely to be negative). This is why variances are smaller in the 2-trait model during most of the year. However, one source of variance is not accounted for in these equations: vertical mixing. Trait variance is not a conservative tracer. Indeed, mixing two communities with different mean trait values "creates" additional variance. As phytoplankton adapts better to their environment in the 2-trait model than in the 1-trait models,

the difference between surface and sub-surface communities when the water column is stratified is larger in the 2-trait model,





and therefore the late Autumn mixing event adds more trait variance in the 2-trait model. This is why the variances are higher in the 2-trait model than in the 1-trait models in December.

## 4  Discussion

### 4.1  Strengths and weaknesses of aggregate models

SPEAD is an aggregate phytoplankton model. Aggregate models, used as far as we know since  Wirtz and Eckhardt (1996), do not compute the abundance of each phytoplankton species as discrete entities, but represent the phytoplankton community by its total biomass together with the mean values, variances and covariances of a few key traits controlling its competitive ability along different environmental gradients. Aggregate models are known to reduce the computational cost of ecosystem models by at least one order of magnitude. In a 0D physical setting (i.e. not spatially resolved), the single-trait aggregate model

of  Acevedo-Trejos et al. (2016) was found to be 18 times faster to run than its alternative discrete model with as few as 10 phenotypes. SPEAD 1.0 is 70 times faster in its aggregate version than in its alternative discrete version with $25 \times 25 = 625$ phenotypes, despite the two model versions having to compute the same number of non-phytoplankton variables. The exact factor of cost reduction depends on the number of phenotypes used in the discrete model, and it is likely to be even larger if the number of traits increases. The computational cost of a multi-phenotype model with phenotypes covering the entire trait domain

is an exponential function of the number of traits. For instance, allowing 25 values of optimal irradiance would multiply the cost of our discrete model by 25. By contrast, the cost of an aggregate model is a quadratic function of the number of traits. In our case, adding a third trait would simply increase the number of phytoplankton state variables from 6 to 10 (total phytoplankton concentration, 3 mean traits, 3 trait variances, 3 inter-trait covariances) and add a few new terms in each equation, which is far less computationally demanding. This makes aggregate models promising tools to explore high-dimensional trait spaces of

the ecology and evolution of microbial ecotypes  (Vallina et al., 2019). We note that some discrete models use an approach different from the one described in our study: they run several simulations with a relatively low number of randomly sampled phenotypes, and then make an ensemble mean of all their simulations. For instance, in  Follows et al. (2007), each member of the ensemble contains 78 random phenotypes, which is not a particularly large number given that they have 4 functional types and that their traits include half-saturation constants for several nutrients, optimal temperature, and optimal irradiance.

Still, running 10 such simulations to compute an ensemble mean is computationally expensive. Aggregate models do not need ensemble means for sampling the trait space since they are continuous by design: their phytoplankton communities fill the trait space completely, without the need of any arbitrary sampling.

Aggregate models are very efficient because their state variables are the quantities that make most ecological sense. In thermodynamics, computing the trajectory of each atom or molecule is not only unfeasible, but also of little use. The collection of

trajectories does not provide more information on the macroscopic behavior of a thermodynamic system than aggregate properties such as temperature, pressure and density. Equally, modeling the dynamics of thousands of species would be incredibly costly, and sufficient observational data would not be available to validate the models. Furthermore, the results would also be extremely difficult to interpret  (Levins, 1966). Given that the most important quantities for understanding a community of





species with similar niches are biomass, followed by mean trait values and trait diversity, the aggregate model focuses computational power where it is most needed, without much loss of information. Aggregate models also explicitly quantify the factors controlling biodiversity, such as the second derivatives of the net growth rate and the trait diffusivity parameters (Chen et al., 2019).

However, the aggregate approach has one major weakness: a specific shape for the trait distribution must be assumed *a priori*, with only as many degrees of freedom as there are free parameters (Wirtz and Eckhardt, 1996; Bruggeman and Kooijman, 2007). There is no universal distribution shape for phytoplankton traits, which is why the equations describing their dynamics are not as precise as the equations of thermodynamics. In this study, we assumed that optimal temperature was normally distributed and that half-saturation constant was lognormally distributed. This Gaussian closure was chosen because of its
simple moment equations and low number of free or arbitrary parameters. Normally distributed temperature and irradiance niches have been observed by Irwin et al. (2015). Half-saturation constant is strongly correlated to cell size (Litchman et al., 2007; Edwards et al., 2012), and lognormal distributions have been observed in nature for size (Cermeño and Figueiras, 2008; Quintana et al., 2008; Schartau et al., 2010; Downing et al., 2014; Marañón, 2015), although not in all cases. Another very common distribution is the power (or "log-linear") law (Rodríguez, 1994; Cermeño and Figueiras, 2008; Huete-Ortega et al.,
2012), although the power law must be truncated on at least one side. We note that a power law distribution with a cutoff on the left might be better able to represent the right-skewed size distribution (even in logarithmic scale) of oligotrophic environments where *Prochlorococcus*, the smallest known phytoplankton, dominates and coexists only with larger species (Marañón, 2015). What neither a unimodal nor a power law distribution can capture is bimodality, which is known to occur at least in lakes, due to common herbivores, in particular daphnids, feeding optimally on preys of intermediate size (Gaedke and Klauschies, 2017).

Normal distributions are symmetrical, unimodal and unbounded. If the real trait distribution deviates from these three properties, errors will arise in aggregate models based on a normal distribution. Not only is information lacking by not including higher-order moments such as skewness and kurtosis, but the dynamics of mean traits and trait variances could be significantly altered. If the trait distribution is skewed, the community will respond faster to a certain type of perturbation than to the opposite perturbation. For instance, if optimal temperature is right-skewed, the phytoplankton community will adapt faster
to warming than to cooling environmental conditions. Phytoplankton with larger optimal temperature will need less time to become dominant in case of warming than cold-water phenotypes in case of cooling because they will start from a higher concentration. To express the above in terms of moment equations, the variance of a right-skewed distribution increases when the environment favors larger trait values, thus facilitating the adaptation, but decreases when the environment favors smaller trait values (see Appendix B and the neglected term $M_{30}$ in equation B7). If the trait distribution is multimodal, the reduction
in trait diversity induced by competitive exclusion (Hardin, 1960) will be slower. This is because replacing all pre-existing communities by intermediate and previously rare phenotypes takes more time than making the most abundant phenotype even more abundant and the rarest even rarer, as in a unimodal distribution. (see Appendix B and the neglected kurtosis term $M_{40}$ in equation B7, knowing that multimodal distributions have low kurtosis).

Normal distributions are unbounded, with the assumption that extreme values are rare and ecologically meaningless. The
consequence of this apparently reasonable assumption is that model phytoplankton can adapt to any environmental change if





they are given enough time, irrespective of the intensity of that change. This contrasts with expectations on the behavior of real phytoplankton communities, as explained in the following example. If a closed (i.e. without immigration) phytoplankton community experiences temperatures between 15 °C and 25 °C, the local phenotypes should be adapted to temperatures between 15 °C and 25 °C and not a single individual should be optimized for temperatures out of the boundaries, since it

would be outcompeted at all places and times. If the environment suddenly warms, the phenotypes with an optimal temperature closer to 25°C should come to dominate. However, if temperature reaches 30°C, no phenotype with an optimal temperature of 30°C can rapidly come to dominate, since no such phenotype pre-exist in the system. Adaptation to temperatures larger than 25°C can only occur through mutations or immigration. In an aggregate model, however, none of these processes are required. Phytoplankton will be able to adapt to any warming because an extremely small but non-zero biomass of phenotypes adapted

to very high temperatures is always present by model design and can become dominant if the environment selects them.

In the present study, the aggregate (continuous-trait) model agrees very well with a multi-phenotype (discrete-trait) model, where no distribution shape is imposed but trait distribution is spontaneously close to normality during most of the year. Skewness and kurtosis occur during some times of the year, only to be removed later, and do not strongly impact our estimates of the lower order moments. The assumed normal trait distribution is symmetrical and unimodal, therefore some errors occur

when the trait distribution is skewed or bimodal. The mean optimal temperature is slightly underestimated and its variance is overestimated because SPEAD does not account for the slight right skew of optimal temperatures distributions. The other main error is that variance tends to decrease too fast in Winter, after the remixing of the previously stratified water column, because SPEAD cannot account for bimodality. The seasonal cycle and the orders of magnitude, however, are accurate. Our results are similar to that of  Acevedo-Trejos et al. (2016). However, other studies show much larger errors  (Coutinho et al., 2016;

Klauschies et al., 2018) and consider Gaussian-based aggregate models to be inaccurate.

Whether the trait distribution of a model ecosystem is normal or not depends on the ecological processes included by the modeler. At least two factors in SPEAD play in favor of a normal distribution. The first factor is trait diffusion. In a fluid, the diffusive movement of a tracer follows a Gaussian law, provided that the diffusivity coefficient is constant  (Einstein, 1905). Trait diffusion plays the same role here for traits and tends to erase skewness and bimodality. The second factor is the simplicity

of our ecological model. All our phytoplankton phenotypes compete for the same resource. In a given environment defined by nutrient concentration and temperature, there is a single most competitive phenotype and the phytoplankton net growth rate decreases continuously when moving away from this optimum. Grazing and mortality rates do not act against this trend because we impose them to be identical for all phenotypes. Conversely, two factors in SPEAD play against normality, but their reach is relatively minor. The asymmetry of the temperature response curve promotes right-skewed distributions of optimal

temperature. However, and despite the naive expectation, this right-skew is generally not large in the discrete model because the standard deviation of optimal temperature is always smaller than the temperature tolerance ($\Delta T = 5$ °C). The second effect is the alternation of stratification and mixing during the year. In Summer, stratification leads to the formation of two distinct communities in surface and in subsurface. When the vertical mixing strengthen again in late Autumn, the two communities mix into a temporary bimodal distribution.

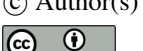

Other ecological settings yield more widespread multimodality. Multimodality can be induced by immigration (Norberg et al., 2001), resting stages (Beckmann et al., 2019), fast environmental oscillations (Beckmann et al., 2019), spatially heterogeneous environments (Wickman et al., 2019), "convex" trade-offs favoring extreme phenotypes (Coutinho et al., 2016), and zooplankton prey selectivity (Wirtz, 2013; Klauschies et al., 2018). In particular, evolutionary branching of both phytoplankton

and zooplankton into tens of size clusters can occur when each zooplankton grazes only on a small size range (Sauterey et al., 2017). However, it is important to point out that only fixed zooplankton preferences cause disruptive selection. By contrast, active switching by grazers ("Kill The Winner") is insufficient to promote evolutionary branching as by design it promotes uniform distributions, flattening the peaks and filling the gaps in trait distributions. Using a Gaussian-based aggregate model in a setting promoting branching would of course be inappropriate: SPEAD would be unable to simulate evolutionary branching

of phenotypes in these kind of ecological scenarios

Alternatives to Gaussian closures have been proposed since early in the development of aggregate models. Norberg et al. (2001) and Norberg (2004) used more complex closures to estimate skewness and kurtosis. However, these closures had free parameters, varying from ecosystem to ecosystem, and a discrete model was required to compute them, canceling the advantage of aggregate models in terms of computational cost. Klauschies et al. (2018) replaced the normal distribution

by a beta distribution. The beta distribution is bounded, allows bimodality, and was proven to increase the realism of trait-based aggregate models in a bounded trait scenario. However, applying this method requires defining fixed boundaries for phytoplankton traits. Phytoplankton has a minimum size (and half-saturation) at 0.5 $\mu$m, which is the size of *Prochlorococcus*, but it does not have a well-defined maximum. Also, optimal temperature at local scales does not have clear boundaries either. Therefore, any set of boundaries would be arbitrary and might prevent further adaptation to changing environments beyond

those limits.

A more practical approach to account for non-Gaussian distributions would be to divide the community into several functional groups, each one having a normal trait distribution of its own (Terseleer et al., 2014; Chen and Laws, 2017). The sum of these communities can have a skewed or multimodal trait distribution. Trait variances (in particular size variance) could be high within phytoplankton as a whole, without bolstering the adaptive capacity of each functional group (see 4.3.). All phenotypes

within a given functional groups must feed on the same nutrients, be subject to the same trade-offs and should not be subject to processes promoting evolutionary branching. Ideally, and in order to prevent the convergence of all functional groups on the same trait values, each group should have distinct qualitative properties or trade-offs. Functional groups could include diatoms, mixotrophs, diazotrophs or *Prochlorococcus*, among others. Two communities defined by the same parameters could coexist and avoid merging if an intermediate trait range is permanently disadvantageous, due for instance to a convex trade-off or to

a size-specific grazer. The multi-Gaussian approach would combine the moderate computational cost of Gaussian aggregate models with the more thorough description of planktonic ecosystems allowed by discrete models.

### 4.2 Traits in phytoplankton community models

In nature, many different traits define phytoplankton niches: nitrogen, phosphorus or iron uptake abilities, requirements in other nutrients (for instance silica or calcite), stoichiometry, optimal temperature, optimal irradiance, mixotrophy, diazotrophy,



motility, buoyancy, resistance to predation, toxicity, and many others. In many trait-based models, this complexity is reduced to one trait. The most common trait is cell size (Terseleer et al., 2014; Acevedo-Trejos et al., 2016; Smith et al., 2016; Chen and Smith, 2018). Cell size is used as a master trait because it is the most observable trait and correlates strongly with many other phytoplankton traits, such as light requirements (Taguchi, 1976; Edwards et al., 2015; Álvarez et al., 2017) and resistance

to predation (Kiørboe, 1993; Thingstad et al., 2005). Some other commonly modeled traits include resistance to predation (Norberg et al., 2001; Merico et al., 2009), optimal temperature (Norberg et al., 2012; Beckmann et al., 2019) and optimal irradiance (Follows and Dutkiewicz, 2011).

The first trait included in SPEAD, half-saturation for nutrients, is known to be strongly correlated to cell size (Litchman et al., 2007; Edwards et al., 2012). Small species, such as cyanobacteria, have low half-saturation constants and can thrive in

oligotrophic waters. They correspond to the "gleaners" of our model. Large species are more likely to be involved in blooms but require more nutrients. They correspond to the "opportunists" of our model. In this study, we chose not to use size but to impose only a simple gleaner-opportunist trade-off. This choice was made for two reasons: to maintain compatibility with the Darwin model (Follows et al., 2007; Vallina et al., 2014a) and to facilitate the analysis of the outputs of our otherwise complex model, as in our setting the best competitors in a given environment have a half-saturation equal to the dissolved inorganic

nitrogen concentration. Our modeling framework can be, however, easily adapted to use size instead of half-saturation as the first trait (Smith et al., 2016).

The trait dynamics of models with 2 traits differ from those of simpler and less realistic single-trait models. Savage et al. (2007) obtain larger trait variances and much larger adaptive capacities when two traits are modeled together rather than in separate models. In our study, we also find a larger adaptive capacity, although it is conveyed by inter-trait correlation only.

We actually find decreased variances, caused by stronger competition, during most of the year, except during the late Autumn re-mixing of the water column, following the Summer stratification. This discrepancy might have been caused by the presence of an immigration term to sustain variance in Savage et al. (2007) but not in SPEAD, since our re-mixing of the water column is most analogous to a dispersal or migration process and we simulate it explicitly. SPEAD simulations coupled with a realistic 3D circulation model where phytoplankton is explicitly allowed to migrate in all directions (ideally in a patchy environment)

would finally tell us if variance is increased or decreased by including more traits.

The low computational costs of aggregate models allows increasing the number of modeled traits, provided that sufficient observational data are available to constrain the corresponding trade-offs. Since the environmental drivers, such as nutrient concentrations, temperature, and light are correlated with each other, the traits are likely to be correlated, unless some processes erasing the correlations are introduced. Regardless of the effect of interactions between traits on variance, multi-trait models

will be able to adapt to their environments faster without the need for large and unrealistic mutation rates or other terms sustaining large variances, such as immigration or Kill The Winner grazing.

### 4.3 Trait diffusion, variance and evolution

Trait diffusion is a key process in SPEAD. Indeed, SPEAD is the first model to include diffusion of multiple traits, providing insights into how both mutations and selection can impact phytoplankton communities. For each modeled trait, a diffusivity





parameter, or "mutation rate", has to be set. The chosen values of these parameters decisively affect trait dynamics. However, the mutation rates remain poorly constrained. The most appropriate rate depends on what the modeler intends to represent by trait diffusion. To understand why, we will need to discuss the notions of "ecological" and "evolutionary" timescales, as well as the notions of "adaptation" and "species".

A first interpretation of trait diffusion is that it is the most conservative way to add variance when the exact processes sustaining trait variance are unknown or too complex to be implemented in models. Indeed, trait diffusion simply adds new variance (aggregate approach) or disperse phytoplankton in a trait space (discrete approach), leaving little room to arbitrary parameters. By contrast, immigration (Norberg et al., 2001) requires assumptions about both immigration rates and the composition of immigrant communities? Likewise, Kill The Winner (Vallina et al., 2014b) requires assumption about whether it is mediated by

viruses or zooplankton, whether consumers are specialized or can actively switch their preference between preys, and whether or not there is a time lag in their response.

    This intepretation of trait diffusion as a generic source of variance is implicitly followed when the diffusivity parameter is set by an optimization algorithm in order to account for the observed trait variance and no other mechanism sustaining variance is included. This way, Chen and Smith (2018) found a diffusivity for the logarithm of size of 0.1, equal to our largest diffusivity

for the logarithm of half-saturation. Even in the homogeneous environments of mesocosms, Wirtz (2013) reports a logarithmic size variance of 0.2 to 0.5, which corresponds to standard deviations between 0.45 and 0.7. In SPEAD, reaching this high values of trait variance is only possible with diffusivities superior or equal to 0.003, despite the fact that our physical setting creates its own variance by mixing phenotypes adapted to the environmental conditions of different depths. This interpretation of trait diffusion is also coherent with the use of trait diffusion as a "variance treatment" (Chen et al., 2019) to study the effect

of diversity on primary production and with the original goal of trait diffusion, which was to sustain trait variance in 0D settings (Merico et al., 2014; Acevedo-Trejos et al., 2016). In these models, trait diffusion is never run combined with other variance-sustaining mechanisms. In real ecosystems, however, mutations are expected to occur at the same time as migration, Kill The Winner grazing, and many other mechanisms that may promote diversity, including multiple convex trade-offs (Beardmore et al., 2011) and mixotrophy (Ward and Follows, 2016).

The way trait diffusion is derived opens a second interpretation of what it represents. Trait diffusion is symmetrical: mutations occur at the same rate toward higher and lower trait values. Trait diffusion is also heritable: mutants transfer their mutations to their offspring. These properties correspond to the evolutionary process of random mutations and selection of the fittest by the environment. It does not correspond to environmentally induced non-heritable variations such as phenotypic plasticity (Ghalambor et al., 2007) or to any selective ecological process driven by the environment, even if some promote

variance.

    According to Fussmann et al. (2007), "Evolution is the change of genotype frequencies within populations or species, whereas community dynamics represent the change of abundances of different species". This definition depends on the notion of "species". Like many phytoplankton models, SPEAD lacks the notion of species and does not distinguish between intraspecific and interspecific trait variance. Our trait space is continuous by design. By mutating, phytoplankton can cross the boundaries between phenotypes, as if they were all of the same species. The only distinction in SPEAD is between mu-





tations, an evolutionary process represented by trait diffusion, and selection, necessary to both ecological (interspecific) and evolutionary (intraspecific) processes and represented by an adaptive change in the mean traits and a decrease in trait variance. Some cases of adaptive evolution to environmental changes in only a few generations have been reported (Fussmann et al., 2007; Kinnison and Hairston, 2007) but are not necessarily caused by mutations occurring at these timescales. They can also be driven by ecological selection on a previously existing intraspecific diversity. In order to choose correct mutation rates these two evolutionary processes must be distinguished.

An alternative interpretation is that ecological processes are particular cases of eco-evolutionary processes where the phenotypes of the offspring are identical to that of their parents (Doebeli et al., 2017). This definition emphasizes the lack of fundamental difference between the two types of processes from a mechanistic point of view, since both are rooted on the same birth-death dynamics. Under this alternative interpretation, the "ecological" timescales are simply the timescales at which the effect of mutations is small and the "evolutionary" timescales are those at which the effect of mutations is large. Therefore, the ecological selection timescales overlap with the adaptive evolution timescales and the difference between the two is diffuse at intermediate timescales. Schlüter et al. (2016) showed that an algal culture starting with a single clone of the abundant coccolithophore *Emiliania huxleyi* could evolve new traits in response to ocean acidification in a time measurable in the laboratory. Their experiment lasted for 2100 generations and the changes after only 100 generations were small. These numbers agree with the seminal studies on *Escherichia coli* where bacteria were shown to adapt to temperature increases or to changes in nutrient availability in 100 to a few thousand generations (Bennett et al., 1990; Lenski et al., 1991; Travisano et al., 1995). These are the timescales a "mutation rate" should reflect. In the case of phytoplankton this means a few years, which falls under the category of "contemporary evolution" but does not allow each species to adapt easily to a seasonal cycle or to faster perturbations. The corresponding trait diffusivity parameters in our study are in the middle or our range, between $3 \times 10^{-5}$ and $10^{-3}$ for half-saturation and between $3 \times 10^{-4}$ and $10^{-2}$ °C$^2$ for optimal temperature.

The trait distributions in SPEAD provide additional insights. In absence of trait diffusion, the two traits become almost totally correlated: one cannot vary without the other. This is a soft version of the diversity collapse observed in 0D models of a 1-trait fitness lanscape. In our 1D model of a 2-trait fitness landscape, trait variances do not collapse to zero but a bidimensional trait space becomes unidimensional and phytoplankton lose their ability to adapt in other directions. Nutrient and temperature niches are known to be correlated in nature, but their correlation is never perfect (Irwin et al., 2012). A small trait diffusivity is sufficient to avoid the collapse of the dimensional trait space into a unidimensional one and likely limits such trait correlations in natural ecosystems, given that mutations affecting half-saturation and optimal temperature are likely independent and hence able to freely fill the full trait space. With very fast trait diffusion, the mean phenotypes adapt instantaneously to their environment but at the cost of keeping a large pool of maladapted phenotypes, which regularly represent more than 15% of the community even if they have very low fitness, because mutations are continuously creating them. These maladapted phenotypes explain the decrease, by up to 10%, of the modeled annual primary production. In nature, the optimal niches of species do not cover all the variability of nutrient concentration and water temperature (Irwin et al., 2012). Furthermore, most real mutations having an effect on the phenotype are deleterious (Timofeeff-Ressovsky, 1940). A mutation rate able to permanently





sustain maladapted phenotypes despite strong selection against them would imply large amount of deleterious mutations not represented in our model, and hence massive mortality. Moderate mutation rates are therefore more likely.

Modeling several communities with their own trait distributions and their own mutations might relax the contradiction between the use of trait diffusion to explain trait variance and the use of trait diffusion to represent evolutionary processes. The variance within a species or a group is lower than the total community variance, and can be sustained with lower mutation rates. This approach can also be used to separate the adaptive evolution of each species from the ecological successions (i.e. inter-group competition) in response to environmental change (Norberg et al., 2012), finally disentangling all components of eco-evolutionary processes.

## 4.4 Future directions

SPEAD 1.0 is the first step of the SPEAD project, whose aim is to simulate plankton evolution with adaptive dynamics in the ocean. In this first version of the model, we kept the complexity manageable, with only one spatial dimension (the vertical) and two physiological traits, in order to facilitate the validation of our aggregate approach and to diagnose the effect of trait diffusion. Three axes of potential future improvement have already been identified: 1) coupling SPEAD with a general circulation model, 2) increasing the number of traits and 3) dividing the community into several functional groups, which implies combining the continuous trait distribution approach with the discrete ecotypes approach.

More concretely, our goal for the near future is to include optimal solar irradiance as a third physiological trait and implement the aggregate approach with trait diffusion in a 3D trait space into the Darwin model (Follows et al., 2007; Dutkiewicz et al., 2009; Barton et al., 2010; Follows and Dutkiewicz, 2011; Ward et al., 2012; Dutkiewicz et al., 2013). Darwin is a versatile model that allows many discrete ecotypes to be resolved along several environmental axes and can be coupled with the MIT general circulation model (Marshall et al., 1997). Optimal irradiance has been present as a trait in the Darwin model since its origin. Indeed, light can be a limiting resource for phytoplankton in the ocean, both in mixed water columns, where plankton cannot stay close to the surface, and near the deep chlorophyll maximum of stratified water columns. Optimal irradiances are known to cover more than two orders of magnitude (Edwards et al., 2015) and to determine the niches of many ecotypes, even within the same species (Biller et al., 2015). The inclusion of optimal solar irradiance as a mutating trait in SPEAD will be a key step to better capture ecological successions. One challenge will be to allow phenotypes to adapt to their environment while accounting at the same time for the mechanistic correlation between nutrient and irradiance niches, since both are related to cell size. The phytoplankton in Darwin were originally divided into four functional groups: *Prochlorococcus* analogs, other small phytoplankton, diatoms and other large phytoplankton. Representing each functional group by its own normal distribution is a possible starting point to develop the multi-Gaussian approach.

Improved versions of SPEAD should be able to address various ecological issues related to community assembly and responses to climate change. By including both trait diffusion and ecological selection of the fittest phenotypes competing in a given environment, SPEAD can potentially be used to disentangle the role of ecological and evolutionary processes in shaping diversity patterns in phytoplankton. In particular, it can be used to determine the conditions under which species or functional groups may survive climate change by evolving new traits or may be replaced by other species or functional groups from other





regions. The effect of environmental changes, such as warming and increased stratification, on plankton size structure, and the effect of biodiversity – controlled by trait diffusion among other processes – on primary production and ecosystem functioning, are other examples of contemporary ecological questions that SPEAD might contribute towards answering.

## 5    Conclusions

In this article, we present an aggregate model of phytoplankton community called SPEAD (Simulating Plankton Evolution with Adaptive Dynamics), where different phenotypes competing for dissolved inorganic nitrogen are characterized by two traits: their half-saturation constants for nitrogen uptake (in logarithmic scale) and their optimal temperature for growth. The phytoplankton community is represented by the six lowest order moments of its trait distribution: total concentration, the mean value of each trait, the variance of each trait, and the inter-trait covariance. The dynamics of these state variables are driven

by three environmental factors: nutrient concentration, temperature, and solar irradiance. The physical setting represents a water column down to 200 m. The seasonal alternation of stratification and vertical mixing also has a strong effect on the trait distribution. Trait diffusion through subsequent generations is included to represent heritable mutations and hence sustain trait diversity. To our knowledge, SPEAD is the first aggregate model to include at the same time two traits (with a proper representation of inter-trait correlation) and trait diffusion.

The ecological parameters of SPEAD were set to reproduce the observed primary production, chlorophyll, nitrate and particulate organic nitrogen concentrations observed by the BATS time series in the Sargasso Sea. Despite its strong assumption that traits are normally distributed, SPEAD was shown to agree precisely with a discrete model explicitly representing all phenotypes, with only minor deviations at depth in Summer, where optimal temperature is underestimated, and in early Winter, where trait variances decrease too fast. This good agreement is made possible by trait diffusion and by the simplicity of our

ecological setting and might not be extendable to all ecosystem models. The trait dynamics depend strongly on the imposed trait diffusivity parameters. With very high diffusivities, primary production is low, variances are high and the two traits are independent, filling the entire trait space. With very low diffusivities, variances are low (albeit non-zero) and the two traits are very strictly correlated: only warm-water gleaners and cold-water opportunists can survive. We think that intermediate values are more realistic, but the precise value depends on whether trait diffusion is meant to sustain the trait diversity of a whole

community or to represent the mutations occurring within a given species.

    SPEAD has a computational cost two orders of magnitudes lower than a full discrete model and its variables are readily interpretable in ecological terms. This effectiveness makes it possible to increase the number of traits. As optimal irradiance is key to explain phytoplankton distribution in the water column and is already present in the Darwin model, the next step of the SPEAD project will be to include it as a third dynamic trait. In agreement with Savage et al. (2007), we showed that

adding traits accelerated the response of preexisting traits to environmental changes. Other venues of future improvement include representing various functional groups, each with their own distinct normal distributions, and coupling SPEAD with a general circulation model. Future versions of our multi-trait framework may address ecological questions related to the impact of selection, mutations, and biodiversity on community dynamics and to the response of phytoplankton to climate change.





## 6  Code and Data availability

The code and data of SPEAD 1.0 are freely available on GitHub (https://github.com/GuillaumeLeGland/SPEAD) and Zenodo (and https://doi.org/10.5281/zenodo.4268431) under the MIT license. The code for SPEAD 1.0 is written in MATLAB (version R2010b) and fully compatible with GNU-Octave (version 4.4.1). To be able to install and operate SPEAD, the user should be familiar with MATLAB or GNU-Octave and have the versions mentioned above or more recent ones. The execution has been tested on Windows with a 2.5 GHz Intel i5-3210M processor, on Linux Ubuntu with a 2.4 GHz Intel Xeon E5645 processor, and on Linux Debian with a 2.6 GHz Intel Xeon E5-2640 processor. The main code modules are:

– SPEAD_1D is the main script to launch SPEAD, calling all functions.

– SPEAD_1D_keys is the function where the different options are declared.

– SPEAD_1D_parameters where the values of the model parameters are assigned.

– SPEAD_gaussecomodel1D_ode45eqs is a function called at each time step to solve the ordinary differential equations of the aggregate (continuous) model.

– SPEAD_discretemodel1D_ode45eqs is a function called at each time step to solve the ordinary differential equations of the multi-phenotype (discrete) model.

SPEAD 1.0 also contains numerous other functions to plot figures and to represent each physical or ecological process (vertical mixing, aggregate trait diffusion, discrete trait diffusion ...). The 4 observations files (for primary production, chlorophyll concentration, nitrate concentration and particulate organic nitrogen concentration) and the 4 external forcing files (for water temperature, surface PAR, vertical mixing and mixed layer depth) are located in the INPUTS folder. Once all files are loaded, SPEAD is run simply by calling SPEAD_1D.

## Appendix A:  Why mutations can be represented as "Trait Diffusion"

In this study, we represented phytoplankton mutations as a "trait diffusion" term, following the work of Merico et al. (2014). In this appendix we show how the expression for trait diffusion is derived and discuss its conditions of validity.

Let us consider the dynamics of a phytoplankton community, where each individual is characterized by the values of two traits, called "x" (in trait unit x or "tux") and "y" (in tuy) . Trait is distributed with a density $p(x,y,t)$ (in mmolN.m$^{-3}$.tux$^{-1}$.tuy$^{-1}$). The mass concentration of phytoplankton cells with values of the first trait between $x$ and $x+dx$ and values of the second trait between $y$ and $y+dy$ is $p(x,y,t) \cdot dxdy$ if $dx$ and $dy$ are small.





If traits are strictly inherited, the equation governing $p(x,y,t)$ for a given phenotype $(x,y)$ depends on the reproduction ($u(x,y,t)$, in $\mathrm{d}^{-1}$) and death ($d(x,y,t)$, in $\mathrm{d}^{-1}$) rates:

$$\frac{\partial p}{\partial t}(x,y,t) = (u(x,y,t) - d(x,y,t))p(x,y,t)$$

$$\frac{\partial p}{\partial t}(x,y,t) = a(x,y,t)p(x,y,t)$$

In the above equation, $a(x,y,t) = u(x,y,t) - d(x,y,t)$ is the net growth rate. In our study, the reproduction rate is identical to the nitrogen uptake rate because nitrogen, the limiting nutrient, is not exuded, cell size is considered independent of time and all nitrogen taken up is used for reproduction. However, genetic mutations or phenotypic plasticity can produce offspring with trait values different from that of their parents. For simplicity, we will consider that mutations increasing or decreasing the traits are equally probable. We assume that the offspring of a parent with trait value $x$ will have trait value $x - \delta_x$ with a

probability $a_x = \nu_x(\delta x)^2$ and trait value $x - \delta_x$ with the same probability, where $\nu_x$ is a diffusivity parameter, expressed in $tux^2$, considered independent of trait value, mutation step ($\delta_x$) and time. Mutations also occur on trait $y$, with a mutation step $\delta y$ and a y-diffusivity parameter $\nu_y$. Mutations on both traits are assumed independent. The probability of having a mutation on both $x$ and $y$ is just the product of the probabilities of each mutation. Hence the time derivative of $p(x,y,t)$ with mutations is:

$$\begin{aligned}
\frac{\partial p}{\partial t}(x,y,t) =\ & [(1-2a_x)(1-2a_y)u(x,y,t) - d(x,y,t)]\,p(x,y,t) \\
& + (1-2a_y)a_x\,[u(x-\delta x,y,t)p(x-\delta x,y,t) + u(x+\delta x,y,t)p(x+\delta x,y,t)] \\
& + (1-2a_x)a_y\,[u(x,y-\delta y,t)p(x,y-\delta y,t) + u(x,y+\delta y,t)p(x,y+\delta y,t)] \\
& + a_x a_y[u(x-\delta x,y-\delta y,t)p(x-\delta x,y-\delta y,t) + u(x+\delta x,y-\delta y,t)p(x+\delta x,y-\delta y,t) \\
& + u(x-\delta x,y+\delta y,t)p(x-\delta x,y+\delta y,t) + u(x+\delta x,y+\delta y,t)p(x+\delta x,y+\delta y,t)]
\end{aligned}$$

In the limit of small but frequent mutations, this equation can be simplified by making a second-order approximation of $u \cdot p$.

$$\begin{aligned}
\frac{\partial p}{\partial t}(x,y,t) =\ & [(1-2a_x)(1-2a_y)u(x,y,t) - d(x,y,t)]\,p(x,y,t) \\
& + (1-2a_y)a_x\left[\left(u \cdot p - \delta_x\frac{\partial(u \cdot p)}{\partial x} + \frac{1}{2}\delta_x^2\frac{\partial^2(u \cdot p)}{\partial x^2}\right) + \left(u \cdot p + \delta_x\frac{\partial(u \cdot p)}{\partial x} + \frac{1}{2}\delta_x^2\frac{\partial^2(u \cdot p)}{\partial x^2}\right)\right] \\
& + (1-2a_x)a_y\left[\left(u \cdot p - \delta_y\frac{\partial(u \cdot p)}{\partial y} + \frac{1}{2}\delta_y^2\frac{\partial^2(u \cdot p)}{\partial y^2}\right) + \left(u \cdot p + \delta_y\frac{\partial(u \cdot p)}{\partial y} + \frac{1}{2}\delta_y^2\frac{\partial^2(u \cdot p)}{\partial y^2}\right)\right] \\
& + a_x a_y\left[u \cdot p - \delta_x\frac{\partial(u \cdot p)}{\partial x} - \delta_y\frac{\partial(u \cdot p)}{\partial y} + \frac{1}{2}\delta_x^2\frac{\partial^2(u \cdot p)}{\partial x^2} + \frac{1}{2}\delta_y^2\frac{\partial^2(u \cdot p)}{\partial y^2} + \delta_x\delta_y\frac{\partial^2(u \cdot p)}{\partial x\partial y}\right] \\
& + a_x a_y\left[u \cdot p + \delta_x\frac{\partial(u \cdot p)}{\partial x} - \delta_y\frac{\partial(u \cdot p)}{\partial y} + \frac{1}{2}\delta_x^2\frac{\partial^2(u \cdot p)}{\partial x^2} + \frac{1}{2}\delta_y^2\frac{\partial^2(u \cdot p)}{\partial y^2} - \delta_x\delta_y\frac{\partial^2(u \cdot p)}{\partial x\partial y}\right] \\
& + a_x a_y\left[u \cdot p - \delta_x\frac{\partial(u \cdot p)}{\partial x} + \delta_y\frac{\partial(u \cdot p)}{\partial y} + \frac{1}{2}\delta_x^2\frac{\partial^2(u \cdot p)}{\partial x^2} + \frac{1}{2}\delta_y^2\frac{\partial^2(u \cdot p)}{\partial y^2} - \delta_x\delta_y\frac{\partial^2(u \cdot p)}{\partial x\partial y}\right] \\
& + a_x a_y\left[u \cdot p + \delta_x\frac{\partial(u \cdot p)}{\partial x} + \delta_y\frac{\partial(u \cdot p)}{\partial y} + \frac{1}{2}\delta_x^2\frac{\partial^2(u \cdot p)}{\partial x^2} + \frac{1}{2}\delta_y^2\frac{\partial^2(u \cdot p)}{\partial y^2} + \delta_x\delta_y\frac{\partial^2(u \cdot p)}{\partial x\partial y}\right]
\end{aligned}$$





The sum of terms in $\frac{\partial(u\cdot p)}{\partial x}$, $\frac{\partial(u\cdot p)}{\partial y}$ and $\frac{\partial(u\cdot p)}{\partial x \partial y}$ is zero, and the sum of factors before $u\cdot p$ is 1. hence the above equation simplifies to:

$$\frac{\partial p}{\partial t}(x,y,t) = [u(x,y,t) - d(x,y,t)]p(x,y,t) + a_x\delta_x^2\frac{\partial^2(u\cdot p)}{\partial x^2} + a_y\delta_y^2\frac{\partial^2(u\cdot p)}{\partial y^2}$$

$$= a(x,y,t)p(x,y,t) + \nu_x\frac{\partial^2(u\cdot p)}{\partial x^2} + \nu_y\frac{\partial^2(u\cdot p)}{\partial y^2}$$

This second-order approximation is valid in the limit as mutations become small ($\delta_x$ and $\delta_y$ tend to zero) and frequent ($a_x = \nu_x(\delta x)^{-2}$ and $a_y = \nu_y(\delta y)^{-2}$ tend to infinity), so that higher-order terms can be neglected. The mathematical expression of the mutation term $\left(\nu_x\frac{\partial^2(u\cdot p)}{\partial x^2} + \nu_y\frac{\partial^2(u\cdot p)}{\partial y^2}\right)$ is somewhat analogous to that representing the diffusion of a tracer in physical

space. In this analogy, the trait space replaces the physical space, $\nu_x\cdot u$ and $\nu_y\cdot u$ replace the diffusivity, and the density $p(x,y,t)$ is the diffused tracer. This analogy is the origin of the phrase "trait diffusion". We note that in a multi-trait space, whatever the number of traits, the diffusion of each trait has the same expression as in the one-trait space.

## Appendix B: Differential equations of a multi-trait aggregate model with trait diffusion

Phytoplankton community models can be discrete or aggregate. In a discrete model, the phytoplankton community is divided into a finite number of phenotypes, each characterized by a different set of trait values. Mutations are discrete with steps equal
to the difference between a phenotype and its nearest neighbors. The differential equation for a discrete phenotype is intuitive and depends only on its net growth rate and a trait diffusion term.

The variables of aggregate models and the differential equations they follow are less intuitive. In an aggregate model, a general shape must be assumed for the trait distribution, with some degrees of freedom, and the prognostic variables are the moments of the trait distribution that are free to vary. In a single-trait model, the most commonly assumed shape is the normal
(or "Gaussian") distribution (Wirtz and Eckhardt, 1996; Bruggeman and Kooijman, 2007), where the phytoplankton density $p(x,t)$ (in $\text{mmolN m}^{-3}\,\text{tux}^{-1}$, with "tux" the "trait unit x") is equal to:

$$p(x,t) = \frac{P(t)}{\sqrt{2\pi V_x(t)}}e^{-\frac{(x-\overline{x}(t))^2}{2V_x(t)}} \tag{B1}$$

In this distribution, the three free parameters are the phytoplankton concentration $P(t)$, the mean trait value $\overline{x}(t)$ and the trait variance $V_x(t)$. They are respectively equal to $\int p(x,t)\cdot dx$, $\int x\cdot p(x,t)\cdot dx$ and $\int(x-\overline{x})^2 p(x,t)\cdot dx$. In a multi-trait space,
we will assume that the traits follow a multivariate normal distribution, which is a generalization of a normal distribution. If N is the number of dimensions:

$$p(x,t) = \frac{P(t)}{(2\pi)^{\frac{N}{2}}det(\Sigma)^{\frac{1}{2}}}e^{-\frac{1}{2}(\boldsymbol{x}-\overline{\boldsymbol{x}}(t))^T\boldsymbol{\Sigma}(t)^{-1}(\boldsymbol{x}-\overline{\boldsymbol{x}}(t))} \tag{B2}$$

In this case, $\boldsymbol{x}$ is the vector containing all traits, $\overline{\boldsymbol{x}}(t)$ is the vector of mean trait values and $\boldsymbol{\Sigma}(t)$ is the matrix of variances and covariances. There are $\frac{(N+1)(N+2)}{2}$ free parameters in total: 1 phytoplankton concentration, $N$ mean trait values, $N$ trait
variances and $\frac{N(N-1)}{2}$ covariances. In the following, we will show how to derive the differential equations for each type of





variable. To this end, we use the method developed by Norberg et al. (2001), based on Taylor expansions of the rates of reproduction and net growth. The assumption of normal trait distribution is only required to compute the time derivatives of variance and covariance, but not for the equations of total biomass and mean trait values.

The trait space is considered to be unbounded, with the implicit assumption that extreme values are extremely rare and ecologically meaningless. This is expressed in the fact that $p$ and all products including $p$ or any of its derivatives tend to 0 when a trait tends to (plus or minus) infinity. For simplicity, in the following part of this section, we will not show the dependencies on the environmental factors and will limit ourselves to two traits, but our method can be extended to derive the equations for any given number of traits. The reproduction and net growth rates of the phenotype defined by trait values $(x,y)$ are denoted $u(x,y)$ and $a(x,y)$ respectively. Integrals are over the whole bi-dimensional domain. We will first derive the equations in the absence of trait diffusion and then discuss what terms are added by the trait diffusion scheme. The net growth of a given phenotype is:

$$\frac{dp}{dt} = a(x,y,t)p(x,y,t) \tag{B3}$$

As a consequence, the equation controlling $P(t)$ is:

$$\frac{dP}{dt} = \int \int a(x,y,t)p(x,y,t) \cdot dxdy$$

We will use the notations $a_{jk}$ (in $d^{-1}.tux^{-j}.tuy^{-k}$) for $a(x,y,t)$ derivated $j$ times with respect to $x$ and $k$ times with respect to $y$, normalized by the factorials of $j$ and $k$, and $M_{jk}$ (in $tux^j.tuy^k$) for the central moment of order $j$ with respect to $x$ and $k$ with respect to $y$:

$$a_{jk}(t) = \frac{1}{j!k!}\frac{\partial^{j+k}a}{\partial x^j \partial y^k}(\overline{x},\overline{y},t)$$

$$M_{jk}(t) = \frac{1}{P(t)}\int \int (x-\overline{x}(t))^j \cdot (y-\overline{y}(t))^k \cdot p(x,y,t) \cdot dxdy$$

We note that $M_{00} = 1$ (by definition of the total concentration), $M_{10} = 0$, $M_{01} = 0$ (by definition of the mean traits), $M_{20}(t) = V_x(t)$, $M_{02}(t) = V_y(t)$ (by definition of the variance) and $M_{11}(t) = C_{xy}(t)$.

A Taylor expansion of the net growth rate on $x$ and $y$ around $(\overline{x}(t),\overline{y}(t))$ yields:

$$
\begin{aligned}
a(x,y,t) &= \sum_{j=0}^{\infty}\sum_{k=0}^{\infty}\frac{1}{j!k!}\frac{\partial^{j+k}a}{\partial x^j \partial y^k}(\overline{x}(t),\overline{y}(t),t) \cdot (x-\overline{x}(t))^j \cdot (y-\overline{y}(t))^k \\
&= \sum_{j=0}^{\infty}\sum_{k=0}^{\infty}a_{jk}(t) \cdot (x-\overline{x}(t))^j \cdot (y-\overline{y}(t))^k
\end{aligned}
$$





20    The time derivative of $P(t)$ depends on the time derivative of $p(x,t)$. Unless explicitly indicated otherwise, all derivatives
with respect to traits are taken at the current mean trait values.

$$
\begin{aligned}
\frac{dP}{dt} &= \int\int \frac{\partial p}{\partial t}(x,y,t)\cdot dxdy \\
&= \int\int a(x,y,t)p(x,y,t)\cdot dxdy \\
&= \int\int \sum_{j=0}^{\infty}\sum_{k=0}^{\infty} a_{jk}(t)\cdot(x-\overline{x}(t))^{j}\cdot(y-\overline{y}(t))^{k}\cdot p(x,y,t)\cdot dxdy \\
&= \sum_{j=0}^{\infty}\sum_{k=0}^{\infty} a_{jk}(t)\int\int (x-\overline{x}(t))^{j}\cdot(y-\overline{y}(t))^{k}\cdot p(x,y,t)\cdot dxdy \\
&= P(t)\sum_{j=0}^{\infty}\sum_{k=0}^{\infty} a_{jk}(t)M_{jk}(t)
\end{aligned}
$$

The first and largest term of this sum is the net growth rate at the mean trait values, denoted $a(\overline{x}(t),\overline{y}(t),t)$. This is the
expected growth rate of a community without trait variance. Since $M_{10}=0$ and $M_{01}=0$, there is no term depending on $\frac{\partial a}{\partial x}$ or
$\frac{\partial a}{\partial y}$. As we want to estimate the effect of trait diversity on the community, we consider the second order terms, proportional to
$V_x(t)$, $V_y(t)$ or $C_{xy}(t)$. Higher order terms, which vanish when variance is small, are neglected, so that:

$$
\frac{dP}{dt} = P(t)\left(a(\overline{x}(t),\overline{y}(t),t) + \frac{1}{2}V_x(t)\frac{\partial^2 a}{\partial x^2} + \frac{1}{2}V_y(t)\frac{\partial^2 a}{\partial y^2} + C_{xy}(t)\frac{\partial^2 a}{\partial x\partial y}\right) \tag{B4}
$$

Second order derivatives are expected to be negative if $(\overline{x}(t),\overline{y}(t))$ is in the neighborhood of the optimal trait value, and
$V_x(t)$ and $V_y(t)$ are always positive. Therefore, the second order terms are generally negative. This means that having large
5    trait variances, or in other words having a large proportion of cells with non-optimal trait values, has a negative effect on the
net community growth.

In the equation for mean trait however, having a large variance is an advantage. Let us define an intermediate variable $S_x(t)$
(in mmolN.m$^{-3}$.tux) as $P(t)\overline{x}(t)$ or, equivalently, as $\int\int x\cdot p(x,y,t)\cdot dxdy$. The time derivative of $S_x(t)$ is:

$$
\begin{aligned}
\frac{dS_x}{dt} &= \int\int x\frac{\partial p}{\partial t}(x,y,t)\cdot dxdy \\
&= \int\int x\cdot p(x,y,t)a(x,y,t)\cdot dxdy \\
&= \int\int (x-\overline{x}(t))\cdot p(x,y,t)a(x,y,t)\cdot dxdy + \overline{x}(t)\int\int p(x,y,t)a(x,y,t)\cdot dxdy \\
&= \int\int \sum_{j=0}^{\infty}\sum_{k=0}^{\infty} a_{jk}(t)\cdot(x-\overline{x}(t))^{j+1}\cdot(y-\overline{y}(t))^{k}\cdot p(x,y,t)\cdot dxdy + \frac{dP}{dt}\overline{x}(t) \\
&= \sum_{j=0}^{\infty}\sum_{k=0}^{\infty} a_{jk}(t)\int\int (x-\overline{x}(t))^{j+1}\cdot(y-\overline{y}(t))^{k}\cdot p(x,y,t)\cdot dxdy + \frac{dP}{dt}\overline{x}(t) \\
&= P(t)\sum_{j=0}^{\infty}\sum_{k=0}^{\infty} a_{jk}(t)M_{j+1,k}(t) + \frac{dP}{dt}\overline{x}(t)
\end{aligned}
$$





15   As $S_x(t)$ is a product, its derivative can also be written as:

$$\frac{dS}{dt} = \frac{d\overline{x}}{dt}P(t) + \frac{dP}{dt}\overline{x}(t)$$

By equating the two previous expressions, we get:

$$\frac{d\overline{x}}{dt} = \sum_{j=0}^{\infty}\sum_{k=0}^{\infty} a_{jk}(t)M_{j+1,k}(t)$$

In this equation, we only consider the highest order terms:

20   $$\boxed{\frac{d\overline{x}}{dt} = V_x(t)\frac{\partial a}{\partial x} + C_{xy}(t)\frac{\partial a}{\partial y}}$$   (B5)

This equation represents the adaptation of the community to its environment. If the mean trait is not optimal, it will increase or decrease in order to maximize the net specific growth rate. The speed of this selection process is proportional to variance : biodiversity is required to track the environmental conditions. The covariance term means that if traits are correlated, the optimization of trait y will also affect trait x.

The mean value of trait y follows a similar equation:

$$\boxed{\frac{d\overline{y}}{dt} = V_y(t)\frac{\partial a}{\partial y} + C_{xy}(t)\frac{\partial a}{\partial x}}$$   (B6)

The equations describing time changes in variances and covariance require more assumptions. As previously, we define an

intermediate variable $Z_x(t)$ (in mmolN.m$^{-3}$.tux$^2$) as $P(t)V_x(t)$ or, equivalently, as $\int\int (x - \overline{x}(t))^2 \cdot p(x,y,t) \cdot dxdy$. In this integral, two terms depend on time: $\overline{x}(t)$ and $p(x,y,t)$. Hence, the time derivative of $Z_x(t)$ is:

$$\frac{dZ_x}{dt} = \int\int (x - \overline{x}(t))^2 \cdot \frac{\partial p}{\partial t}(x,y,t) \cdot dxdy + \int\int \frac{\partial (x - \overline{x}(t))^2}{\partial t}p(x,y,t) \cdot dxdy$$

$$= \int\int (x - \overline{x}(t))^2 \cdot p(x,y,t)a(x,y,t) \cdot dxdy - 2\frac{d\overline{x}}{dt}\int (x - \overline{x}) \cdot p(x,y,t) \cdot dxdy$$

By definition of $\overline{x}(t)$, we have $\int\int (x - \overline{x}) \cdot p(x,y,t) \cdot dxdy = 0$. Thus:

$$\frac{dZ_x}{dt} = P(t)\sum_{j=0}^{\infty}\sum_{k=0}^{\infty} a_{jk}(t)M_{j+2,k}(t)$$

As $Z_x(t)$ is a product, its derivative can also be written as:

$$\frac{dZ_x}{dt} = \frac{dV_x}{dt}P(t) + \frac{dP}{dt}V_x(t)$$

By equating the two previous expressions, we get:

$$\frac{dV_x}{dt} = \frac{1}{P(t)}\left(\frac{dZ_x}{dt} - \frac{dP}{dt}V_x(t)\right)$$

$$= \sum_{j=0}^{\infty}\sum_{k=0}^{\infty} a_{jk}(t)M_{j+2,k}(t) - V_x(t)\sum_{j=0}^{\infty}\sum_{k=0}^{\infty} a_{jk}(t)M_{jk}(t)$$





In this equation, the two terms proportional to $a(\overline{x}(t), \overline{y}(t), t)$ compensate and it is no longer possible to neglect the third and fourth orders of the trait distribution. With only the two lowest order non-zero terms retained, the time derivative of variance is:

$$\frac{dV_x}{dt} = M_{30}(t)\frac{\partial a}{\partial x} + M_{21}(t)\frac{\partial a}{\partial y} + \frac{1}{2}(M_{40}(t) - V_x^2(t))\frac{\partial^2 a}{\partial x^2} + \frac{1}{2}(M_{22} - V_xV_y)\frac{\partial^2 a}{\partial y^2} + (M_{31} - V_xC_{xy})\frac{\partial^2 a}{\partial x \partial y} \qquad \text{(B7)}$$

The moments $M_{30}(t)$ and $M_{40}(t)$ are called the skewness and kurtosis of x respectively. They represent the shape of the trait distribution. These moments could be described by their own equations but their time derivatives depend on moments of even higher orders, and so on. In order to limit mathematical complexity and computation time, we do not explicitly compute moments of higher order than variance. Instead, we close our system by giving these moments the same value as in a bivariate normal distribution. In a bivariate normal distribution, odd order moments (where $j + k$ is odd) are zero (for reasons of

symmetry) and even order moments can be expressed as a function of variances (Isserlis, 1916):

$$M_{40}(t) = 3V_x^2$$
$$M_{31}(t) = 3V_xC_{xy}$$
$$M_{22}(t) = V_xV_y + 2C_{xy}^2$$
$$M_{13}(t) = 3V_yC_{xy}$$
$$M_{04}(t) = 3V_y^2$$

The equation for $V_x(t)$ then simplifies to:

$$\boxed{\frac{dV_x}{dt} = V_x^2\frac{\partial^2 a}{\partial x^2} + 2V_xC_{xy}\frac{\partial^2 a}{\partial x \partial y} + C_{xy}^2\frac{\partial^2 a}{\partial y^2}} \qquad \text{(B8)}$$

The equation for $V_y(t)$ is obtained by swapping the x and y indices:

$$\boxed{\frac{dV_y}{dt} = V_y^2\frac{\partial^2 a}{\partial y^2} + 2V_yC_{xy}\frac{\partial^2 a}{\partial x \partial y} + C_{xy}^2\frac{\partial^2 a}{\partial x^2}} \qquad \text{(B9)}$$

These expressions represent the effect of competition on trait variance. If the mean trait is near an optimum, then trait diversity tends to collapse due to competitive exclusion.

The equation for covariance is derived in a similar way. We define $Z_{xy}(t)$ as $P(t)C_{xy}(t)$. The time derivative of $Z_{xy}(t)$ is:

$$\begin{aligned}
\frac{dZ_{xy}}{dt} &= \int\int (x - \overline{x}(t)) \cdot (y - \overline{y}(t)) \cdot \frac{\partial p}{\partial t}(x,y,t) \cdot dxdy - \frac{d\overline{x}}{dt}\int\int (y - \overline{y}(t)) \cdot p(x,y,t) \cdot dxdy \\
&\quad - \frac{d\overline{y}}{dt}\int\int (x - \overline{x}(t)) \cdot p(x,y,t) \cdot dxdy
\end{aligned}$$

The last two terms are zero since $M_{10} = 0$ and $M_{01} = 0$. Thus:

$$\begin{aligned}
\frac{dZ_{xy}}{dt} &= \int\int (x - \overline{x}(t)) \cdot (y - \overline{y}(t)) \cdot p(x,y,t)a(x,y,t) \cdot dxdy \\
&= P(t)\sum_{j=0}^{\infty}\sum_{k=0}^{\infty} a_{jk}(t)M_{j+1,k+1}(t)
\end{aligned}$$





Since $Z_{xy}$ is a product, its derivative can also be written as:

$$\frac{dZ_{xy}}{dt} = \frac{dC_{xy}}{dt}P(t) + \frac{dP}{dt}C_{xy}(t)$$

By equating the two previous expressions, we get:

$$\begin{aligned}\frac{dC_{xy}}{dt} &= \frac{1}{P(t)}\left(\frac{dZ_{xy}}{dt} - \frac{dP}{dt}C_{xy}(t)\right)\\ &= \sum_{j=0}^{\infty}\sum_{k=0}^{\infty}a_{jk}(t)M_{j+1,k+1}(t) - C_{xy}(t)\sum_{j=0}^{\infty}\sum_{k=0}^{\infty}a_{jk}(t)M_{jk}(t)\end{aligned}$$

In the case of a bivariate normal distribution, the only terms remaining produce the following equation:

$$\frac{dC_{xy}}{dt} = \frac{1}{2}\left[M_{31}(t) - C_{xy}(t)V_x(t)\right]\frac{\partial^2 a}{\partial x^2} + \frac{1}{2}\left[M_{13}(t) - C_{xy}(t)V_y(t)\right]\frac{\partial^2 a}{\partial y^2} + \left[M_{22}(t) - C_{xy}^2(t)\right]\frac{\partial^2 a}{\partial x \partial y}$$

Replacing $M_{31}(t)$, $M_{22}(t)$ and $M_{13}(t)$ by their expressions as a function of lower order moments yields:

$$\frac{dC_{xy}}{dt} = C_{xy}(t)V_x(t)\frac{\partial^2 a}{\partial x^2} + C_{xy}(t)V_y(t)\frac{\partial^2 a}{\partial y^2} + \left[V_x(t)V_y(t) + C_{xy}^2(t)\right]\frac{\partial^2 a}{\partial x \partial y} \tag{B10}$$

Competition tends to reduce variance, which must therefore be sustained by another process. In our model, this is the role of trait diffusion, described in Appendix A and originally derived by Merico et al. (2014). Trait diffusion represents the effect of heritable mutations on the trait distribution and adds two new terms to the net growth equation (B3) that are mathematically very similar to tracer diffusion:

$$\frac{dp}{dt} = a(x,y,t)p(x,y,t) + \nu_x\frac{\partial^2(u \cdot p)}{\partial x^2}(x,y,t) + \nu_y\frac{\partial^2(u \cdot p)}{\partial y^2}(x,y,t) \tag{B11}$$

The additional term for the total biomass time derivative due to the diffusion of $x$ is:

$$\int\int \nu_x\frac{\partial^2(u \cdot p)}{\partial x^2} \cdot dxdy = \nu_x\int\left[\frac{\partial(u \cdot p)}{\partial x}\right]_{-\infty}^{+\infty}dxdy = 0$$

This result comes from the fact that $u \cdot p$ and its derivatives tend toward zero when any trait tends toward infinity. The effect of the diffusion of $y$ would equally be zero. This means trait diffusion is not a source of biomass. The additional term for $\frac{dS_x}{dt}$

due to the diffusion of x is:

$$\begin{aligned}\int\int \nu_x \cdot x\frac{\partial^2(u \cdot p)}{\partial x^2} \cdot dxdy &= \nu_x\int\left[x \cdot \frac{\partial(u \cdot p)}{\partial x}\right]_{-\infty}^{+\infty}dy - \nu_x\int\int\frac{\partial(u \cdot p)}{\partial x} \cdot dxdy\\ &= \nu_x\int\left[x \cdot \frac{\partial(u \cdot p)}{\partial x}\right]_{-\infty}^{+\infty}dy - \nu_x\int\left[u \cdot p\right]_{-\infty}^{+\infty}dy\\ &= 0\end{aligned}$$

The effect of $y$ diffusion on $\overline{x}$ and of both $x$ and $y$ diffusion on $\overline{y}$ are equally zero. Trait diffusion has no effect on the mean

trait equations. It does not favor any direction of change. However, trait diffusion is a source of variance. Indeed, a similar




integration by parts adds a new non-zero term to $\frac{dZ_x}{dt}$:

$$\int\int \nu_x \cdot (x - \overline{x}(t))^2 \cdot \frac{\partial^2(u \cdot p)}{\partial x^2}(x,y,t) \cdot dxdy$$

$$= \int \nu_x \left[ (x - \overline{x}(t))^2 \cdot \frac{\partial(u \cdot p)}{\partial x} \right]_{-\infty}^{+\infty} dy - 2\nu_x \int\int (x - \overline{x}(t)) \cdot \frac{\partial(u \cdot p)}{\partial x}(x,y,t) \cdot dxdy$$

$$= 0 - 2\nu_x \int [(x - \overline{x}(t)) \cdot u \cdot p]_{-\infty}^{+\infty} dy + 2\nu_x \int\int u(x,y,t)p(x,y,t) \cdot dxdy$$

$$= 2\nu_x \int\int u(x,y,t)p(x,y,t) \cdot dxdy$$

This integral is similar to $\frac{dP}{dt}$ (replacing $a$ by $u$), and can be Taylor-expanded in the same way. The contribution of $y$ diffusion on $V_x$ is a different case. The extra term on $\frac{dZ_x}{dt}$ is zero:

$$\int\int \nu_y \cdot (x - \overline{x}(t))^2 \cdot \frac{\partial^2(u \cdot p)}{\partial y^2}(x,y,t) \cdot dxdy$$

$$= \nu_y \int (x - \overline{x}(t))^2 \left[ \int \frac{\partial^2(u \cdot p)}{\partial y^2}(x,y,t) \cdot dy \right] dx$$

$$= 0$$

Thus, the new equations of $V_x(t)$ and $V_y(t)$ accounting for trait diffusion are:

$$\frac{dV_x}{dt} = V_x^2 \frac{\partial^2 a}{\partial x^2} + 2V_x C_{xy} \frac{\partial^2 a}{\partial x \partial y} + C_{xy}^2 \frac{\partial^2 a}{\partial y^2} + 2\nu_x \left[ u(\overline{x}(t), \overline{y}(t), t) + \frac{1}{2}V_x(t)\frac{\partial^2 u}{\partial x^2} + \frac{1}{2}V_y(t)\frac{\partial^2 u}{\partial y^2} + C_{xy}(t)\frac{\partial^2 u}{\partial x \partial y} \right] \quad (B12)$$

$$\frac{dV_y}{dt} = V_y^2 \frac{\partial^2 a}{\partial y^2} + 2V_y C_{xy} \frac{\partial^2 a}{\partial x \partial y} + C_{xy}^2 \frac{\partial^2 a}{\partial x^2} + 2\nu_y \left[ u(\overline{x}(t), \overline{y}(t), t) + \frac{1}{2}V_x(t)\frac{\partial^2 u}{\partial x^2} + \frac{1}{2}V_y(t)\frac{\partial^2 u}{\partial y^2} + C_{xy}(t)\frac{\partial^2 u}{\partial x \partial y} \right] \quad (B13)$$

Because $u(x,y,t)$ is positive for every value of $x$ and $y$, the trait diffusion term is positive. Trait diffusion is able to counter the effect of competitive exclusion and sustain variance.

The effect on $Z_{xy}(t)$ is:

$$\int\int \nu_x (x - \overline{x}(t)) \cdot (y - \overline{y}(t)) \cdot \frac{\partial^2(r \cdot c)}{\partial x^2}(x,y,t) \cdot dxdy$$

$$= \nu_x \int (y - \overline{y}(t)) \left[ \int (x - \overline{x}(t)) \cdot \frac{\partial^2(r \cdot c)}{\partial x^2}(x,y,t) \cdot dx \right] dy$$

$$= 0$$

Thus trait diffusion does not add nor remove covariance to the phytoplankton community. However, by increasing the variances, trait diffusion decreases the correlation. In other words, trait diffusion decorrelates the traits by making rare trait combinations more likely.

We note that, in absence of trait diffusion, our equations are a particular case of the general equations derived by Bruggeman (2009) for multivariate normal trait distributions. In a single trait case, they are simpler than the original equations (Merico et al., 2014) but identical to the more recent formulation of Coutinho et al. (2016) and derived using the same method.





*Author contributions.* GL, SMV, SLS and PC conceived and designed the study. SMV wrote the initial code for 1 trait and acquired the observational data. GL derived the equations for 2 traits and the trait diffusion scheme, built the final code and wrote the first draft. All authors contributed to and reviewed the manuscript.

*Competing interests.* The authors declare that they have no conflict of interest.

*Acknowledgements.* The authors thank all the scientists who produced the data and developed all the concepts used in this article. In particular, we thank Agostino Merico, Esteban Acevedo-Trejos and Marcel Oliver for the discussions we had on trait diffusion. This work was funded by the national research grant CTM2017-87227-P (SPEAD) from the Spanish government. We acknowledge support of the publication fee by the CSIC Open Access Publication Support Initiative through its Unit of Information Resources for Research (URICI).





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
