# Peer review of "SPEAD 1.0 – A model for Simulating Plankton Evolution with Adaptive Dynamics in a two-trait continuous fitness landscape applied to the Sargasso Sea"

_Geoscientific Model Development, 2020_

## Referee Comment (RC1) · Fanny Monteiro (Referee) · 30 Dec 2020

The authors present the development of a new evolutionary model of phytoplankton (SPEAD) that represents for the first time the mutation of two distinctive traits (half-saturation constant and optimal temperature) in a continuous trait space. The primary question is to see if we can develop a computationally faster model of plankton diversity that accounts for trait evolution using a Trait Diffusion term (aggregate models). The authors couple their model to a 1D vertical water column to compare their results with seasonal observations at the station BATS. They also compare the new model

with a "discrete" version. This already tested modelling approach represents diversity with hundreds of different phytoplankton types (sampling the trait space available in the evolutionary model). Once the new model is validated, the authors explore the role of mutation rate on the diversity dynamics as this is a critical term that is poorly constrained. The primary outcomes are: (1) Despite its strong assumption that traits are normally distributed, SPEAD is shown to agree precisely with the observations and a discrete model explicitly representing all phenotypes, with only minor deviations at depth in Summer, where the optimal temperature is underestimated, and in early Winter, where trait variances decrease too fast; (2) The trait dynamics depends strongly on the imposed trait diffusivity parameters; (3) SPEAD has a computational cost two orders of magnitudes lower than a discrete model, making aggregate models promising tools to explore high-dimensional trait spaces of the ecology and evolution of microbial ecotypes. The model's code is also available online on two open sources with instructions on getting the model running.

I think this is a critical work for evolutionary marine biology and ecosystem modelling. It is a step forward testing for the first time the simultaneous mutation of more than one trait, which requires the inclusion of the covariance between traits. The authors carefully explain, validate and test their new approach. Besides, this work provides a detailed overview of the history of modelling diversity and evolution, some in-depth discussion on the meaning of their method, its strengths and weaknesses, and some ideas of what could be done differently to improve their work (a multiple Gaussian approach). I strongly favour the publication of this work in GMD. It provides the necessary step toward computationally fast evolutionary modelling with an in-depth analysis of their new modelling approach's implications and unknowns.

Main comments:

Overall, the manuscript lacks a brief discussion (ideally in the introduction and discussion) on what evolutionary models bring over the traditional diversity functional types models (like Darwin). What are the processes missing in the PFT models that evolu-

tionary models get and in which environments are these processes essential? Including such a discussion will help make the paper less niche by bringing the non-evolution community up-to-speed with your approach.

The introduction provides a really detailed and useful review of the development of evolution modelling for phytoplankton. Can I suggest that you make a diagram showing the different approaches with their assumptions, cost and benefits? This would help to see how your work fits in with previous model developments.

P23, line 7: You need to justify why the ratio of mutation rates are assumed to be the same and discuss the implication on your model results. It seems that might significantly affect your results - see p26, line 7 and line 23.

Specific comments:

Abstract (p1, l2-3): Here, you highlight the main challenge of modelling evolution, I don't think that is the only one. Can you amend your sentence to reflect this?

In general, the abstract lacks a definition of your primary question that the work tackles, which you do very well in the manuscript. Can you include it in the abstract?

Introduction P2, 1st line: Some phytoplankton can be multicellular. Please amend here.

P2, line 10: The production of DMS by marine plankton is now highly debated. Can you reflect this here?

P3, lines 14-16: Can you define clearly and quite early on in the introduction the term "aggregate model"? You refer to it in line 8 (continuously distributed traits) but do not link the two. This is confusing and key for the paper.

The Methods section needs some clarifications:

P9, lines 33-35: Here you assume that $f p u \infty p$ is independent of x, with x the half-saturation constant but below you define $K_n = u \infty p / f p$. Can you clarify this point? It seems that x depends on $f p u \infty p$ because $K_n = x = 1/(fp)2 \, f p u \infty p$.

Equation 15 does not seem consistent as it assumes that Kn(x)=e(x). Is that correct?

Equation 16: why did you define the maximum nutrient limitation with a squared root?

P11, line 7: Here you define, in my mind, a critical assumption that probably has a significant impact on your result: fpu∞p is a constant. Can you add more discussion on this assumption (where it comes from and its limitation and influence on your results)?

P14, lines 11-14: The definition for "a" is too vague. Can you provide an equation of "a" with its correction terms and describe the meaning behind them?

P14, lines 16-17: Is that based on the theory, on what your equations predict or on your model's results? Please amend.

P19, line 22: should be observed (rather than "observated").

Section 3.1 could be more concise if the model's values and observations were presented in a table.

Figure 6: please zoom in for the x and y axis as currently too small.

P34, lines 8-9: Not a question. Please rephrase.

P35, lines 1-2: This is an essential point but not the most straightforward sentence (it is too long). Can you amend to make this information more readable?

P37, line 5: replace "an aggregate model" with "a new aggregate model".

P37, line 23: Modify to include "intermediate values of mutation rates (or diffusivity).

―――――――――――――――――

---

## Referee Comment (RC2) · Anonymous Referee #2 · 1 Feb 2021

Review to "SPEAD 1.0 – A model for Simulating Plankton Evolution with Adaptive Dynamics in a two-trait continuous fitness landscape applied to the Sargasso Sea"

This study nicely demonstrates that an aggregate model may capture the key properties of functional diversity within phytoplankton communities, i.e. the means, the variances and the covariance of two physiologically important traits, over the course of a year very well, while being at the same time superior with respect to the computational efficiency when compared to a corresponding discrete multispecies model. This implies that aggregate models might be a suitable tool to study the impact of functional

diversity on the population and community dynamics of natural plankton communities, in particular when facing a multidimensional trait space.

Despite my overall positive impression of the present study, I have the feeling that the study would benefit from more clarity in the method section where the model is initially described. Furthermore, I am not entirely convinced at this point to what extent the addition of functional diversity to the model may improve the match between the modeled and observed phytoplankton densities and nutrient concentrations over a spatial gradient and over the course of a year. For details, please see below.

Major

1) The model validation is done based on e.g. the observed primary production and chlorophyll concentration of the phytoplankton in the Sargasso Sea. Overall, the model predictions match the observations quite well. However, I was wondering whether a corresponding model that only uses a mean field approach and thus neglects trait diversity and trait diffusion entirely, performs really worse when compared to the performances of the aggregate and discrete-diversity models. I mean, the sensitivity analysis performed with respect to the comparison of the 2D and two different 1D models shows that the predicted temporal development of the phytoplankton density is quite similar throughout most times of the year. Based on this, I was wondering how these models differ from a model that does not account for trait diversity at all, and thus a 0D model with respect to the functional diversity. This additional model simulation may help to answer the question whether a model that accounts for trait diversity matches the observation better than a model which does not. Furthermore, this consideration may also help to support the claim that additional measurements of functional diversity of natural communities in respect to relevant traits are needed to actually validate aggregate models, which account for trait diversity.

2) The uptake function up is introduced twice, i.e. with equation (11) and equation (12). The authors used equation (12) to consider two different limiting cases of the specific nitrogen uptake rate up, i.e. equations (13) and (14). Based on this, the authors motivated their choices of how the half saturation constant and maximum uptake rate should depend on the trait x given the constraint, that the species' abilities to take up nitrogen at low and high concentrations should follow a gleaner-opportunist trade-off. However, after having defined the uptake function up with equation (11), I would have found it more intuitive to proceed with the definition of the nutrient limiting factor ÏŠ, which is simply assumed to follow Michalis Menten kinetics with a corresponding half saturation constant and maximum uptake rate. Afterwards, with or without a limit consideration, the gleaner-opportunist trade-off could have been motivated by stating that the maximum uptake rate to the power of 2 multiplied by the half saturation constant is assumed to remain constant. Please consider a potential revision of this section!

3) The sign structure of the migration or diffusion terms in equation (20) seems to be mixed up. I mean, the outflow of density from the focal population should be associated with a minus sign whereas the inflow of density towards the focal population should correspond to positive signs.

4) In section 2.3 Physical setting, the authors describe that vertical mixing is included via vertical (physical) diffusion. However, I could not really understand how the addition of vertical diffusion have modified the equations (30) to (35) that are governing the biomass and trait dynamics of the phytoplankton community. So, please explain in a little bit more detail how this implicit scheme, which is mentioned on page 15, lines 15-18 actually works and how the corresponding terms or equations are looking like. Accordingly, please also add the terms that are reflecting sinking.

Minor

Page 4, Lines 14/15: This statement about a future perspective seems somewhat misplaced in the introduction.

Page 5, equation (1): Is the uptake function Up that uses a capital letter U the same as the uptake function up that uses a small letter u and which is introduced later on in the

text (e.g. equation (10))? While Up seems to depend on P, up does not depend on it. Please clarify?

Page 21, Line 2: Please add a "by" between "are caused" and "the aggregate model's assumption".

Page 24, Line 16: Please replace "is" by "are", i.e. "the standard deviations ... are significantly smaller".

Page 24, Line 20: Here, it says that Table 3 also assesses whether bimodality occurs at some moment in time. However, I was wondering how exactly this was done by the authors. I could not find any explanation. Please clarify!

Page 34, Line 9: Please substitute the "?" by a ".".

Page 34, Line 12: Spelling error: "interpretation" not "intepretation".

Page 35, Line 21: Please substitute "or" by "of".

Page 39, Line 1: Why do you mention here that cell size is independent of time? Cell size is not a trait, that you have explicitly incorporated into your model. Please clarify.

Page 39, Line 4: Please substitute the minus sign '-' by a plus sign '+' in the term x-$\delta$x.

Page 39, Line 5: The probability ax should be a ratio instead of a product and thus read ax= vx/($\delta$x^2) instead of vxÂů($\delta$x^2). Otherwise the term ($\delta$x^2) would not cancel out in the equation that is displayed in line 29 of page 40. See also line 28 of page 40 where ax and ay are displayed correctly.

---

## Author Comment (AC1) · 1 Mar 2021

[bg, manuscript]copernicus

[Figure]

**gmd-2020-302**

SPEAD 1.0 – A model for Simulating Plankton Evolution with Adaptive Dynamics in a two-trait continuous fitness landscape applied to the Sargasso Sea

Guillaume Le Gland, Sergio M. Vallina, S. Lan Smith, and Pedro Cermeño

**Answers to reviewers**

Dear reviewers,

Thank you for your thoughtful comments on our article. These comments helped us a lot to improve the scientific quality of the manuscript and to make it more accessible to the readers. Please find our answers to your comments below, point by point, colored in blue. Please inform us (legland@icm.csic.es) if you cannot read the text.

Sincerely yours,

The authors

**Reviewer 1 (Fanny Monteiro)**

Main comments: Overall, the manuscript lacks a brief discussion (ideally in the introduction and discussion) on what evolutionary models bring over the traditional diversity functional types models (like Darwin). What are the processes missing in the PFT

models that evolutionary models get and in which environments are these processes essential? Including such a discussion will help make the paper less niche by bringing the non-evolution community up-to-speed with your approach. The introduction provides a really detailed and useful review of the development of evolution modelling for phytoplankton. Can I suggest that you make a diagram showing the different approaches with their assumptions, cost and benefits? This would help to see how your work fits in with previous model developments.

This comment has been very useful, as indeed our introduction, and to some extent our abstract and discussion, have been excessively focused on mechanisms sustaining trait variance. We are now providing a broader content on why eco-evolutionary models are important for marine ecology. A paragraph has been added in the introduction before the presentation of SPEAD to explain what the main interests of 1) the aggregate approach and 2) the presence of evolution in models. A more detailed review of the different types of models with their assumptions, costs and benefits would require an article in its own right. We thus refer to Ward et al. (2019) detailed review on the subject. In section 4.4 of the discussion, called "Future Directions", these issues have been further linked to future venues of research using the SPEAD model, in particular the use of future versions of SPEAD to evaluate plankton adaptation to climate change. The present article is a detailed technical description of a new modeling approach without addressing yet a major scientific question requiring the explicit modeling of evolution.

https://agupubs.onlinelibrary.wiley.com/doi/full/10.1029/2018MS001452

P23, line 7: You need to justify why the ratio of mutation rates are assumed to be the same and discuss the implication on your model results. It seems that might significantly affect your results - see p26, line 7 and line 23.

The ratio of the two mutation rates has been set based on the range of each environmental parameter. There is no reason why it should be exactly $10°C2$ in nature. We do not particularly promote this value. The ratio is an unknown in itself. Simulations with other values of the ratio have been performed, but we did not include them in the article because the impact of the ratio on results is insignificant compared to the overall effect of increasing or decreasing both mutation rates. The fit with the discrete model is neither better nor worse, the total primary production lies in the same range, the dynamics of each tracer is similar to other simulations where it has the same mutation rate, and the inter-trait correlation is somewhere in between the simulations sharing either one or the other mutation rate. We have made the lack of impact of this ratio clearer by amending the first paragraph of section 3.3.

Abstract (p1, l2-3): Here, you highlight the main challenge of modelling evolution, I don't think that is the only one. Can you amend your sentence to reflect this? In general, the abstract lacks a definition of your primary question that the work tackles, which you do very well in the manuscript. Can you include it in the abstract?

Certainly the first paragraph of the abstract was originally focused on explaining the mechanisms that sustain trait variance in competitive communities. We have now re-oriented the abstract to highlight the role of trait diversity as an ecological mechanism leading to adaptive evolution of communities subject to new environmental conditions. We have also added in the second paragraph of the abstract that "The SPEAD model can be used to evaluate plankton adaptation to environmental changes at different timescales or address ecological issues affected by adaptive evolution", which highlight the rationale of this paper: testing a new method that will later be used to answer various questions in theoretical plankton ecology.

Introduction P2, 1st line: Some phytoplankton can be multicellular. Please amend here.

We have replaced "single-cell" by "microscopic" in this sentence and later state that phytoplankton "are mainly single-cell although some colonial or multicellular species exist in most phyla (Beardall et al., 2009)". Certainly some phytoplankton taxa can form colonies and some are even multi-cellular (such as Volvox). However, the SPEAD model does not resolve such level of detail and simulates the eco-evolutionary dynamics of pelagic single-cell phytoplankton phenotypes.

https://nph.onlinelibrary.wiley.com/doi/10.1111/j.1469-8137.2008.02660.x

P2, line 10: The production of DMS by marine plankton is now highly debated. Can you reflect this here?

The production of DMS by marine plankton is well established (Simo 2001, TREE; Levasseur 2013, Nature Geo; Royer et al. 2016, SciRep; Marti Gali et al. 2019, PNAS) although there is a long-standing and ongoing debate regarding the influence of DMS on cloud properties and the Earth's radiative budget (Quinn and Bates 2011, Nature). Given that the biogenic sulfur cycling is not an essential aspect of our article, we have removed any mention to DMS dynamics altogether.

https://www.sciencedirect.com/science/article/abs/pii/S0169534701021528

https://www.pnas.org/content/pnas/116/39/19311.full.pdf

https://www.nature.com/articles/ngeo1910

https://www.nature.com/articles/nature10580

P3, lines 14-16: Can you define clearly and quite early on in the introduction the term "aggregate model"? You refer to it in line 8 (continuously distributed traits) but do not link the two. This is confusing and key for the paper.

We use "aggregate model" and "continuous-trait model" interchangeably. "Aggregate" is more accurate, as our state variables are the "aggregate properties". "Continuous-trait" is by experience simpler to understand and highlights the contrast with discrete models. The names of the models have been made clearer in the fourth paragraph of the introduction (p3). The equivalence between "discrete" and "multi-phenotype" has also been clarified.

P9, lines 33-35: Here you assume that $f_p u_p^\infty$ is independent of x, with x the half-saturation constant but below you define $K_n = \frac{u_p^\infty}{f_p}$. Can you clarify this point? It seems that x depends on $f_p u_p^\infty$ because $K_n = x = \frac{1}{f_p^2} f_p u_p^\infty$.

This section has been thoroughly revised following your comments and those of reviewer 2. $f_p u_p^\infty$ = constant is the expression of the gleaner-opportunist trade-off. In the revised version, $K_n$, the variable that is probably most familiar to the general reader, is introduced first. The formula relating it to the affinity ($f_p$) and the maximum growth rate ($u_p^\infty$) is now derived using a simpler but mathematically equivalent approach. Stating that $K_n = \frac{1}{f_p^2} f_p$ is a valid mathematical manipulation. However, it does not mean that "$K_n$ (or x) depends on $f_p u_p^\infty$", because $f_p u_p^\infty$ is independent of x by design (this is something we impose, we do not prove it). Therefore the only correct conclusion is that $K_n$ is proportional to $f_p^{-2}$, which happens to be true in our model. Yet, to avoid any potential confusion or misunderstanding of our approach, we have simplified the description of the equations.

Equation 15 does not seem consistent as it assumes that Kn(x)=e(x). Is that correct?

Thank you for reporting this typo. In the denominator, $K_n(x)$ is not equal to $e^x$ but to $K_n^0 * e^x$, for reasons of dimensions. Exponentials and logarithms do not have dimensions, half-saturation constants do. This correction restores consistence with the previous definition of x.

[Figure]

Equation 16: why did you define the maximum nutrient limitation with a squared root?

As stated above, this section has been thoroughly revised following your comments and those of reviewer 2. We now explain the mathematical manipulations leading to this square root expression. $f_p u_p^\infty$ = constant (gleaner-opportunist trade-off) and $f_p = \frac{u_p^\infty}{K_n}$ (as the relation between the three quantities is now defined) lead to $\frac{\left(u_p^\infty\right)^2}{K_n}$ = constant, and thus to the fact that $u_p^\infty$ is proportional to the square root of $K_n$.

P11, line 7: Here you define, in my mind, a critical assumption that probably has a significant impact on your result: $f_p u_p^\infty$ is a constant. Can you add more discussion on this assumption (where it comes from and its limitation and influence on your results)?

As now explained in the text of our revised article: "This trade-off has been experimentally observed in bacteria and phytoplankton (Cermeño et al., 2011; Vallina et al., 2019 and references therein) and has already been used in models (Dutkiewicz et al., 2009 GBC; Vallina et al., 2014 ncomms; Smith et al., 2016; Vallina et al., 2017). It is the simplest way to discriminate the bloom-forming opportunist phytoplankton (such as diatoms) from the ubiquitous gleaners (such as Prochlorococcus)."

Although this trade-off has been observed, the mechanisms explaining it are not well known. As stated in Vallina et al. (2019): "One of such trade-offs, experimentally measured in bacteria and phytoplankton (Healey and Hendzel, 1980; Button et al., 2004; Elbing et al., 2004), appears to exist between specific affinity $\alpha_p$ and specific maximum uptake rate $\rho_{max}$ (Litchman et al., 2015; Vallina et al., 2017) such that $\rho_{max}\alpha_p$ = constant, although the thermodynamic basis of this trade-off are yet unclear (Wirtz, 2002)". The root of the trade-off may be "the way uptake sites are packed in the finite cell surface (Aksnes and Egge, 1991)", that is, on the number of transporters, on their affinity for nutrients and on their handling rate.

With $f_p u_p^\infty$ = constant, $K_n$ correspond to the nutrient concentration at which a phenotype outcompetes all others in non-equilibrium conditions. This property makes this

trade-off particularly convenient for a technical study. More complex trade-offs, where the optimal nutrient niche and the half-saturation would be distinct quantities, could bring confusion as their results would be more difficult to analyze.

http://www.int-res.com/articles/meps2011/429/m429p019.pdf

https://www.researchgate.net/publication/334271629_Models_in_Microbial_Ecology

https://agupubs.onlinelibrary.wiley.com/doi/full/10.1029/2008GB003405

https://www.nature.com/articles/srep34170

https://www.sciencedirect.com/science/article/pii/S0304380016303246

http://www.int-res.com/articles/meps/70/m070p065.pdf

P14, lines 11-14: The definition for "a" is too vague. Can you provide an equation of "a" with its correction terms and describe the meaning behind them?

The net growth rate "a" is the sum of the ecological "source and sink" terms affecting the phytoplankton community, such as the growth by nutrient uptake (u), the mortality by grazing (g), and the background natural mortality (m). We have made it clearer in the paragraph after Eqs. 30-35 which explains these equations in non-mathematical terms. The correction terms are those present after "a" in Eq. 30, that is: $1/2 V_x \frac{\partial^2 a}{\partial x^2} + 1/2 V_y \frac{\partial^2 a}{\partial y^2} + C_{xy} \frac{\partial^2 a}{\partial x \partial y}$. The paragraph has also been amended to link every notion to its corresponding mathematical expression more explicitly.

P14, lines 16-17: Is that based on the theory, on what your equations predict or on your model's results? Please amend.

These two lines are interpretations of the equations, not results. I have amended them to link them explicitly to the equations.

P19, line 22: should be observed (rather than "observated").

Thank you. This has been corrected.

Section 3.1 could be more concise if the model's values and observations were presented in a table.

We appreciate the suggestion but in our opinion Figures are more informative than tables. Furthermore, Figure 4 is a 2D (depth,time) plot that cannot be summarized in a simple Table. Section 3.1 is already the most concise result section of the article.

Figure 6: please zoom in for the x and y axis as currently too small.

All panels except biomass (a) have been zoomed in. An error on the half-saturation units in panel b) has also been corrected.

P34, lines 8-9: Not a question. Please rephrase

This has been corrected: The "?" has been replaced by "."

P35, lines 1-2: This is an essential point but not the most straightforward sentence (it is too long). Can you amend to make this information more readable?

If we understand correctly, the long sentence to amend is: "The only distinction in SPEAD is between mutations, an evolutionary process represented by trait diffusion, and selection, necessary to both ecological (interspecific) and evolutionary (intraspecific) processes and represented by an adaptive change in the mean traits and a decrease in trait variance." For the sake of clarity, it has been reformulated in five shorter sentences. A few details were added: "The only distinction in SPEAD is between mutations and selection. Mutations are a strictly evolutionary process. In SPEAD they are represented by trait diffusion through subsequent generations. Selection encompasses

both intraspecific selection, which is a second evolutionary process, and interspecific selection, which is an ecological process. In SPEAD, selection has two effects: it drives the mean traits towards their optimum values (see Eqs. 30-31) and decreases trait variances (see Eqs. 32-34) by eliminating the rarest and least fit phenotypes."

P37, line 5: replace "an aggregate model" with "a new aggregate model".

This sentence has been amended as requested.

P37, line 23: Modify to include "intermediate values of mutation rates (or diffusivity).

This sentence has been amended as requested.

**Reviewer 2**

1) The model validation is done based on e.g. the observed primary production and chlorophyll concentration of the phytoplankton in the Sargasso Sea. Overall, the model predictions match the observations quite well. However, I was wondering whether a corresponding model that only uses a mean field approach and thus neglects trait diversity and trait diffusion entirely, performs really worse when compared to the performances of the aggregate and discrete-diversity models. I mean, the sensitivity analysis performed with respect to the comparison of the 2D and two different 1D models shows that the predicted temporal development of the phytoplankton density is quite similar throughout most times of the year. Based on this, I was wondering how these models differ from a model that does not account for trait diversity at all, and thus a 0D model with respect to the functional diversity. This additional model simulation may help to answer the question whether a model that accounts for trait diversity matches

the observation better than a model which does not. Furthermore, this consideration may also help to support the claim that additional measurements of functional diversity of natural communities in respect to relevant traits are needed to actually validate aggregate models, which account for trait diversity.

We have performed the "0D model with respect to the functional diversity" simulation asked by the reviewer (see attached) and it will be added to the Supplementary Information. This supplement also shows bulk properties with different mutation rates. The differences in bulk properties are smaller than the differences between model and observations. The largest difference between all the simulations is the total primary production but the distribution of all bulk properties in space and time remains very similar. This was to be expected due to the simplicity of our model, where primary production mainly depends on nutrient fluxes and thus on vertical mixing, and where sinking and predation do not depend on traits. In more complex settings, this may not to be the case. The goal of the validation against Sargasso Sea data is not to prove that modeling traits is necessary but to "assess the realism of the trait-independent biogeochemical parameters (Table 2)", as is now explained in section 3.1.

This good and relevant question is essentially related to the long-standing debate about "simple models vs. complex models" regarding trait diversity for simulating marine plankton ecosystems. A simple NPZD model with mean trait values for the single Phytoplankton (Kn, Topt) may provide or not similar simulations of bulk properties as a Darwin type model with many ecotypes, depending on some ad-hoc choices. Simple NPZD can be very valid for simulating bulk properties for global biogeochemical cycles and carbon fluxes at the global scale. But for them to work, they usually impose i) an intermediate Michaelis-Menten uptake curve for nutrient uptake, which is fine; ii) and a single Eppley upper (envelope) curve for temperature dependence without explicitly modelling the individual niches that are underneath; which is a strong ad-hoc assumption. Using an Eppley curve is implicitly assuming many individual niches in temperature that are not explicitly resolved. A single ecotype in nature has an optimal temperature and a tolerance range much smaller than the ad-hoc imposed Eppley curve for temperature dependence. That is, imposing the Eppley curve for temperature dependence is implicitly assuming a multi-species phytoplankton model under a NPZD hood. Alternatively, the NZPD model could have a single explicit temperature niche (not with the full Eppley curve, just one Gaussian niche) for the phytoplankton population but then the simulations will show that primary production only happens during a narrow temporal window: when the environmental conditions match the "optimal temperature and a tolerance range" of the phytoplankton population.

This has been shown in Vallina etal. ECOMOD 2017, where varying the functional diversity ($K_n$, $T_{opt}$) of a phytoplankton community using a NPZD model leads to different values of bulk properties such as primary production. The simplest configuration is equivalent to having a single phytoplankton species (i.e. all ecotypes are equal in trait values) while the more complex configuration is equivalent to having a multiple phytoplankton species (i.e. all ecotypes are different in trait values). The 1024 individual simulations performed that covered all potential combinations in trait-values clearly showed that allowing for trait-variability in the community results in a more productive and stable functioning of the ecosystem. That is, functional trait-diversity of phytoplankton ecotypes leads to what is known as the "insurance hypothesis" of biodiversity. Furthermore, if we want marine ecosystem modelling to go beyond the current global biogeochemical cycles mindset and start answering ecology & evolution questions, we have to include trait-variability and trait-mutations in the next generation of models. A simple NPZD can maybe provide a good fit to bulk properties but it will not be useful to understand eco-evo processes such as the rules of community assembly, competition and cooperation, food web stability, evolutionary branching, etc.

In particular, SPEAD can be used to simulate the adaptive evolution of plankton to projected climate change conditions and evaluate the influence of different mutation rates in the face of the observed climate change velocity (Brito-Morales et al., 2020) SPEAD can assess the relative importance of adaptive evolution, migration and extinction. This cannot be done with a NPZD model. Even primary production and biomass may not be assessed correctly in this case, because parameterizations used for the current climate, such as Kn, have no reason to be valid under a future climate. In the future, eco-evolutionary models may also be used to study speciation after extinction events or the relative roles of neutral evolution and ecological selection in community assembly. These aspects are more developed in the introduction and discussion of our revised version, following a comment of reviewer 1.

https://www.nature.com/articles/s41558-020-0773-5

2) The uptake function up is introduced twice, i.e. with equation (11) and equation (12). The authors used equation (12) to consider two different limiting cases of the specific nitrogen uptake rate up, i.e. equations (13) and (14). Based on this, the authors motivated their choices of how the half saturation constant and maximum uptake rate should depend on the trait x given the constraint, that the species' abilities to take up nitrogen at low and high concentrations should follow a gleaner-opportunist trade-off. However, after having defined the uptake function up with equation (11), I would have found it more intuitive to proceed with the definition of the nutrient limiting factor, which is simply assumed to follow Michalis Menten kinetics with a corresponding half saturation constant and maximum uptake rate. Afterwards, with or without a limit consideration, the gleaner-opportunist trade-off could have been motivated by stating that the maximum uptake rate to the power of 2 multiplied by the half saturation constant is assumed to remain constant. Please consider a potential revision of this section!

This paragraph where the nutrient-dependent growth factor is explained has been revised. The paragraph now begins by stating that the trait x is the logarithm of the half-saturation constant and by presenting the Michaelis-Menten dynamics controlling $\gamma_N$. These are the most simple and familiar concepts. The expressions for the affinity $f_p$ and the maximum growth $u_p^\infty$ rate are then introduced and used to justify the trade-off ($f_p * u_p^\infty$ = constant) as the simplest way to allow competition by giving an ecological

niche in a nutrient concentration gradient to each phenotype. The mathematical remark that $\left(u_p^\infty\right)^2 \left(K_n\right)^{(} - 1)$ is also constant will help us in explaining the expression of $\gamma_N(x)$, which apparently surprised both reviewers. However, $\left(u_p^\infty\right)^2 \left(K_n\right)^{(} - 1) =$ constant is not a good definition for the trade-off because it is not clear on what is traded off against what. $K_n$ does not define the competitive ability in any particular environment. A phenotype with a very high $K_n$, which is normally disadvantageous, can perfectly well dominate in all environments if it has a very high $u_p^\infty$. By contrast, a phenotype with a low affinity is always at a disadvantage at low nutrient concentration. This is because $K_n$ is a composite trait defined as the ratio of two primary traits (nutrient affinity divided by maximum uptake rate) and therefore $K_n$ has two degrees of freedom in the absence of a gleaner-opportunist trade-off. The paragraph now has 4 equations instead of 5. Only the first and last equations are necessary to reproduce the model and are very simple. The other two explain the underlying assumptions behind the trade-offs. This is now made explicit in the main text.

3) The sign structure of the migration or diffusion terms in equation (20) seems to be mixed up. I mean, the outflow of density from the focal population should be associated with a minus sign whereas the inflow of density towards the focal population should correspond to positive signs.

You are right, the signs in the trait-diffusion terms were mixed up. This error has been corrected. Thank you.

4) In section 2.3 Physical setting, the authors describe that vertical mixing is included via vertical (physical) diffusion. However, I could not really understand how the addition of vertical diffusion have modified the equations (30) to (35) that are governing the biomass and trait dynamics of the phytoplankton community. So, please explain in a little bit more detail how this implicit scheme, which is mentioned on page 15, lines 15-18 actually works and how the corresponding terms or equations are looking like.

Accordingly, please also add the terms that are reflecting sinking.

We originally chose not to include transport and sinking in the sections 2.1 and 2.2 in order to be able to focus on biogeochemical processes, which are complex enough. Since you are requesting it, the full equations for all tracers have now been added to section 2.3, with sinking and transport. "Implicit" refers to the scheme employed to estimate the vertical derivatives. The code is available on Github if the reader wants to know more about this implicit scheme. Explicit schemes do exist but they are often numerically unstable unless a very small time step is imposed. In case you were instead referring to the conversions between the state variables $(P, \overline{x}, \overline{y}, V_x, V_y, C_{xy})$ and the conserved moments $\left(P, P\overline{x}, P\overline{y}, P(V_x + \overline{x}^2), P(V_y + \overline{y}^2), P(C_{xy} + \overline{xy})\right)$. In fact, the conserved moments are the tracers used in transport and whose derivatives are used to proceed from one time step to the following. In the new version of section 2.3, we now explain how their derivatives are computed, based on the derivatives of the state variables and on the formula of vertical diffusion. There also were some minor typos in the expressions of the conserved variances and covariance. These typos have been corrected.

Page 4, Lines 14/15: This statement about a future perspective seems somewhat misplaced in the introduction

The statement about a future perspective has been moved to the section 4.4 of the discussion: "Future directions".

Page 5, equation (1): Is the uptake function Up that uses a capital letter $U$ the same as the uptake function up that uses a small letter u and which is introduced later on in the text (e.g. Eq. (10))? While $U_p$ seems to depend on P, $u_p$ does not depend on it. Please clarify?

"$U_p$" with a capital "$U$" is the total nitrogen uptake by the phytoplankton community (in mmolN m$^{-3}$ d$^{-1}$). $U_p$ depends on $P$ (phytoplankton concentration). "$u_p$" with a lower-

case "$u$" is the biomass-specific uptake rate of each phytoplankton phenotype (in d$^{-1}$). It is multiplied by phytoplankton concentration $P$ (discrete case) or density $P_{ij}$ (continuous case) but do not depend on them. "$U_p$" has been rebranded as "$V_p$" to avoid this confusion. The units of up are also mentioned the first time it appears. Before equation (1) I now specify that all the fluxes on the right hand side are in mmolN m$^{-3}$ d$^{-1}$.

Page 21, Line 2: Please add a "by" between "are caused" and "the aggregate model's assumption"

This has been corrected.

Page 24, Line 16: Please replace "is" by "are", i.e. "the standard deviations . . . are significantly smaller".

This sentence has been removed. After correcting a minor error in the code (see below) the control simulation now fully converges and all the comments to explain how specific it is appear useless.

Page 24, Line 20: Here, it says that Table 3 also assesses whether bimodality occurs at some moment in time. However, I was wondering how exactly this was done by the authors. I could not find any explanation. Please clarify!

The following sentence has been added: "Bimodality was assessed visually based on the trait distributions of the discrete model during the December mixing event".

Page 34, Line 9: Please substitute the "?" by a ".".

This typo (also noted by reviewer 1) has been corrected.

Page 34, Line 12: Spelling error: "interpretation" not "intepetation".

This typo has been corrected.

Page 35, Line 21: Please substitute "or" by "of".

This typo has been corrected.

Page 39, Line 1: Why do you mention here that cell size is independent of time? Cell size is not a trait, that you have explicitly incorporated into your model. Please clarify.

Cell size here is not mentioned as a trait. What we meant is that we assume plankton do not use resources to increase their size or biomass but only to reproduce. We have made it clearer in the text. The phrase "cell size is considered independent of time" has been replaced by "cells do not modify their nitrogen content" for more clarity.

Page 39, Line 4: Please substitute the minus sign '-' by a plus sign '+' in the term $x - \delta x$.

This error has been corrected.

Page 39, Line 5: The probability ax should be a ratio instead of a product and thus read $a_x = \frac{\nu_x}{(\delta x2)}$ instead of $\nu_x (\delta x)^2$. Otherwise the term $(\delta x) 2$ would not cancel out in the equation that is displayed in line 29 of page 40. See also line 28 of page 40 where $a_x$ and $a_y$ are displayed correctly

You are perfectly right. Thank you. The exponent +2 has been corrected to -2. It now reads "$a_x = \nu_x (\delta x)^{-2}$" instead of "$a_x = \nu_x (\delta x)^2$".

**Additional changes:**

We discovered an error in the two variance equations of our GNU OCTAVE / MATLAB code. This has been corrected on Github. The Appendix B is correct. In the code, the factor 2 of the second right-hand terms ($2V_x C_{xy} \frac{\partial^2 a}{\partial x \partial y}$ and $2V_y C_{xy} \frac{\partial^2 a}{\partial x \partial y}$) in equations (B12) and (B13) was missing. Since these are minor terms in most simulations, the effects were minimal, much smaller than the discrepancy between the continuous and the discrete models. This is why this computer bug went overlooked. In the figures, differences are only noticeable for very low mutation rates. All simulations have been repeated with this error corrected. Table 3, Figure 8 and Figure 10 have been modified accordingly. This error did not affect the conclusions of our study.